# Abrupt transitions in time series with uncertainties

Bedartha Goswami [1,2], Niklas Boers [1,3], Aljoscha Rheinwalt [1,2], Norbert Marwan [1], Jobst Heitzig[1], Sebastian F.M. Breitenbach[4] & Jürgen Kurths[1,5,6]

Identifying abrupt transitions is a key question in various disciplines. Existing transition detection methods, however, do not rigorously account for time series uncertainties, often neglecting them altogether or assuming them to be independent and qualitatively similar. Here, we introduce a novel approach suited to handle uncertainties by representing the time series as a time-ordered sequence of probability density functions. We show how to detect abrupt transitions in such a sequence using the community structure of networks representing probabilities of recurrence. Using our approach, we detect transitions in global stock indices related to well-known periods of politico-economic volatility. We further uncover transitions in the El Niño-Southern Oscillation which coincide with periods of phase locking with the Pacific Decadal Oscillation. Finally, we provide for the first time an 'uncertainty-aware' framework which validates the hypothesis that ice-rafting events in the North Atlantic during the Holocene were synchronous with a weakened Asian summer monsoon.

[1] Potsdam Institute for Climate Impact Research, Transdisciplinary Concepts & Methods, 14412 Potsdam, Germany. [2] Institute of Earth and Environmental Science, University of Potsdam, Karl-Liebknecht Str. 24-25, 14476 Potsdam, Germany. [3] Grantham Institute - Climate Change and the Environment, Imperial College London, South Kensington Campus, London, SW7 2AZ, UK. [4] Sediment and Isotope Geology, Institute for Geology, Mineralogy & Geophysics, Ruhr-Universität Bochum, Universitätsstr. 150, 44801 Bochum, Germany. [5] Department of Physics, Humboldt-Universität zu Berlin, Newtonstr. 15, 12489 Berlin, Germany. [6] Saratov State University, 83 Astrakhanskaya Street, Saratov 410012, Russia. Niklas Boers and Aljoscha Rheinwalt contributed equally to this work. Correspondence and requests for materials should be addressed to B.G. (email: goswami@pik-potsdam.de)

Time series analysis is an indispensable framework that helps us to understand dynamical systems based on temporally ordered observations[1], and uncertainties should, in principle, form a crucial part of the inferences made from time series. An important question addressed in time series analysis is the identification of abrupt transitions—time points at which the observable suddenly shifts from one type of behavior to another. Transition detection approaches have relevance in a wide range of disciplines such as (paleo)climate[2,3], ecology[4], and finance[5]. Although there exists a vast literature on various kinds of 'change-point detection' methods that seek to address this question[6–10], most of these approaches tend to simplify the nature of uncertainties in the data in exchange for analytical tractability, thereby influencing whether or not a transition is detected in the time series. Here, we claim that the lack of thorough uncertainty propagation stems from the way time series are represented. We thus put forth a new representation of time series that naturally includes its uncertainties and show how it can be used to detect abrupt transitions more reliably.

A 'time series' is typically constructed as an ordered sequence of point-like measurements $\{x_t\}_{t=1}^N$ of an observable $X$. Quantitative methods are thereafter employed to analyze $\{x_t\}_{t=1}^N$ and the propagation of uncertainties (if provided) is carried out as a separate exercise. This makes the error analysis highly nontrivial, perceived often merely as an addition to the core analysis and results, and also allows investigators to ignore or postpone it. Even if an error analysis is performed, the errors are often assumed to be independent and (qualitatively) identical, which is inaccurate for most real-world observables and may lead to substantial pitfalls. Here, we introduce a framework that merges the analysis of the measurements with that of their errors, and shifts the focus from knowing the value of an observable at a given time to knowing how likely it is that the observable had a chosen value at that time.

In contrast to the traditional time series framework, where the time evolution of the state $\mathcal{X}$ of a system is encoded in a series of point-like measurements $\{x_t\}_{t=1}^N$, we propose to represent the time evolution of $\mathcal{X}$ using a sequence of probability density functions (PDFs), $\{\rho_t^X\}_{t=1}^N$, such that $\rho_t^X(x)$ encodes our beliefs about the likelihood of the value $X = x$ at time $T = t$. The state $\mathcal{X}$ is linked to the observable $X$ by a measurement process which may be noisy, and the PDFs encode our partial knowledge about $\mathcal{X}$, which may occur due to measurement imprecision, or due to spatio-temporal fluctuations. For instance, if we consider $\mathcal{X}$ to be the sea surface temperature (SST) anomaly of the Niño 3.4 region in the equatorial Pacific at a given time $t$, we assume, in our framework, the observation $x_t^u$ at grid location $u$ to be a noisy estimate of $\mathcal{X}$. We then use the values $x_t^u$ from all locations $u$ inside the Niño 3.4 box to construct the probability density $\rho_t^X$, which encodes our partial knowledge about the state $\mathcal{X}$ of the SST anomaly of the Niño 3.4 box at time $t$ (Fig. 1). Repeating this for different time instances results in a sequence of PDFs $\{\rho_t^X\}_{t=1}^N$ which encode our partial knowledge about the time evolution of $\mathcal{X}$.

Using $\{\rho_t^X\}_{t=1}^N$ instead of $\{x_t\}_{t=1}^N$ to represent the temporal evolution of the state $\mathcal{X}$ offers several advantages: it explicitly shows how our knowledge of $\mathcal{X}$ might often be encoded by non-Gaussian PDFs that change from one time point to the next. The PDFs of the SST anomalies within the Niño 3.4 region during the 1997–1998 El Niño, for instance, shown in Fig. 1b, are clearly non-normal and keep changing their shape over time. The $\{\rho_t^X\}_{t=1}^N$ representation of the time evolution of $\mathcal{X}$ also reveals implicit (and often inaccurate) assumptions. For example, the classic Pearson's correlation coefficient between two observables is estimable only if we can quantify our knowledge about their joint occurrences at each time instant, or if we can assume them to be independent when conditioned on a given time $T = t$ (see Supplementary Note 1).

The PDFs $\{\rho_t^X\}_{t=1}^N$ can be estimated in different ways depending on the system under study and the nature of the measurements. In this paper, we present three real-world examples from considerably distinct disciplines and the PDF sequences in each of these is constructed in a different way. Irrespective of how the PDFs $\{\rho_t^X\}_{t=1}^N$ are obtained, the main goal of our paper is to demonstrate that $\{\rho_t^X\}_{t=1}^N$ is a viable representation for time series datasets, potentially useful in a wide range of real-world applications, and to show how we can detect abrupt transitions in time series with uncertainties with the help of the PDF sequence.

Our proposed transition detection approach correctly identifies abrupt shifts in global stock indices that correspond to well-known periods of political and economic volatility. We are further able to detect transitions in the El Niño-Southern Oscillation (ENSO), which turn out to be coincident to periods in which the ENSO is phase locked to the Pacific Decadal Oscillation (PDO). Furthermore, in the case of paleoclimatic proxy records from Asia, we provide a clear and principled validation of the hypothesis that North Atlantic ice-rafting episodes during the last 13,000 years were synchronous to periods of weakening of the Asian summer monsoon (ASM). In contrast to most existing approaches for detecting change points in time series, our proposed PDF sequence representation transparently takes into account the time series uncertainties without simplifications, and demonstrates how the identification of transitions are impacted by these uncertainties. Particularly for the paleoclimate records from the ASM domain, inferences related to the spatio-temporal propagation of Holocene cold events are crucially impacted by the magnitude and nature of uncertainties contained in the time series.

## Results

**Detecting abrupt transitions using recurrence network communities**. To detect abrupt transitions, we extend the framework of recurrence analysis[11], a valuable tool to investigate features such as memory, disorder, and synchronization from patterns encoded in the return characteristics of the state $\mathcal{X}$. Traditionally, the first step is to estimate a binary recurrence matrix whose elements indicate (with a 1 or a 0) if a chosen time point recurred to an earlier state or not. Various estimates derived from the recurrence matrix help to quantify the processes that give rise to the temporal evolution of the observable[11]. Here, we use the series $\{\rho_t^X\}_{t=1}^N$ to estimate the probability of recurrence for all pairs of time points in a way that does not require us to assume independence, or to quantify the dependence between the probability distributions at two different time instances (see Methods). We construct an estimator $\hat{\mathbf{A}}$ of the recurrence probabilities (Eq. 14), interpreted as the adjacency matrix of a network whose nodes are the observation time points and whose edge weights are the recurrence probabilities. Such a network obtained from a single time series is referred to as a recurrence network[12] (RN) and has been used to study various complex systems[13–16]. Next, we use the community structure of the RN as an indicator of abrupt transitions: communities[17] in a RN are time intervals with a higher similarity within themselves than to the rest of the time series, indicating a shift in the dynamics near the borders between different communities. We demonstrate this with a synthetic example (Eqs. 1 and 2) where three different transitions are imposed on a noisy sinusoidal signal (Fig. 2). If we consider only the mean time series, we fail to detect the transition at $T = 675$ and can date the other two only much coarser, further highlighting why we should represent time series as $\{\rho_t^X\}_{t=1}^N$ rather than $\{x_t\}_{t=1}^N$.

We apply our method to three real-world examples with three different sources of uncertainty (noted in parentheses): daily financial stock index data from 2004 to 2016 (intra-day temporal variability), monthly SST anomalies from 1881 to 2012 for the

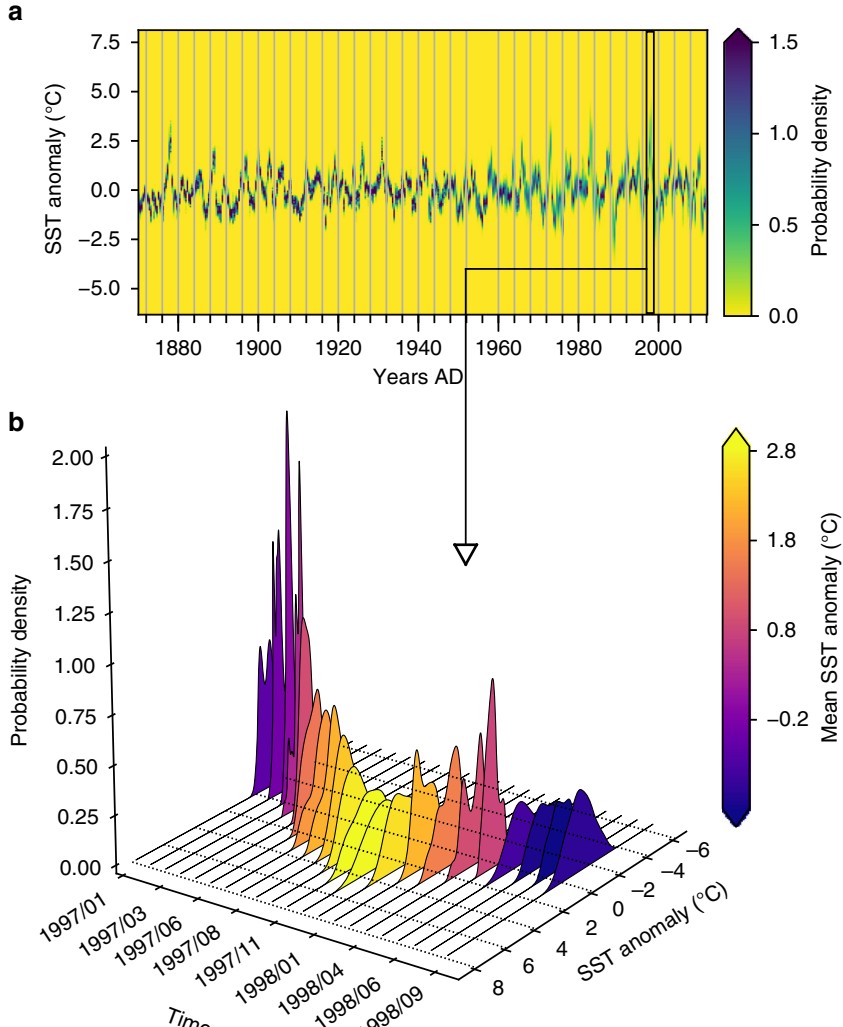

**Fig. 1** Representation of a time series as a sequence of probability density functions. The time series of PDFs $\{\rho_t^X\}_{t=1}^N$ is shown in (**a**) for monthly SST anomalies from the Niño 3.4 region, from late 1870 to 2012. The densities are estimated using a kernel density estimation procedure (see Methods) which gives a probability density of SST anomalies for each month given the spatially distributed measurements. Each vertical column in (**a**) is a density $\rho_t^X$ color coded according to its value when evaluated for different values of the SST observable X. Darker (lighter) colors in each column thus represent higher (lower) chances of observing the corresponding SST anomalies for that month in the Niño 3.4 region. We propose to consider such a series $\{\rho_t^X\}_{t=1}^N$ of PDFs instead of representing them as point estimates. The $\{\rho_t^X\}_{t=1}^N$ sequence is shown in detail using a three-dimensional (3D) representation in (**b**) for the SST anomalies during the 1997–1998 El Niño (black box in (**a**)). The color of each density in (**b**) denotes the average SST anomaly for that month, clearly indicating the Niño-like conditions during the winter of 1997–1998, but we also see the non-Gaussian nature of the probability densities throughout the period, calling into doubt the efficacy of representative point estimates such as the mean

Niño 3.4 region (spatial variability), and paleoclimatic proxy records from Asia covering important intervals of the Holocene (imprecision in determining proxy ages). A $\{\rho_t^X\}_{t=1}^N$ sequence is constructed from the data and used to detect abrupt transitions (Methods, Supplementary Note 2, and Supplementary Figs. 1–3). In each case, we repeat the analysis using only the mean time series, and find that using the probability density sequences gives more reliable and robust detection of abrupt transitions (Supplementary Note 3, and Supplementary Figs. 4–6).

**Transitions in financial stock indices**. First, as a proof of concept of the proposed approach, we consider three stock market indices: DAX (Frankfurt), NASDAQ-100 (New York), and BSE SENSEX (Mumbai). We identify three groups of abrupt transitions (Fig. 3a–c) centered around the 'mortgage crisis', the 'Eurozone crisis', and the 'Brexit'/'Grexit' crises, as indicated by corresponding peaks in the Google trends data (Fig. 3d). The end of 2009 marks a common period of abrupt transitions and instabilities for all three indices during the worst part of the US mortgage crisis, symbolized here by the bankruptcy claim of Lehmann Brothers. Of the two queries 'Grexit' and 'Brexit', we note that the transitions show a better correspondence with the former. Additional events in the BSE SENSEX in 2006 and 2015 coincide with large intra-day falls in the Mumbai-based stock index on 22 May 2006 and 24 August 2015. Abrupt shifts detected in the BSE in May 2014 roughly coincide with the national parliamentary elections held in India that year, in which there was a shift from a decade-old rule by the center-left United Progressive Alliance to the center-right National Democratic Alliance, suggesting a volatile period for the Mumbai-based stock exchange.

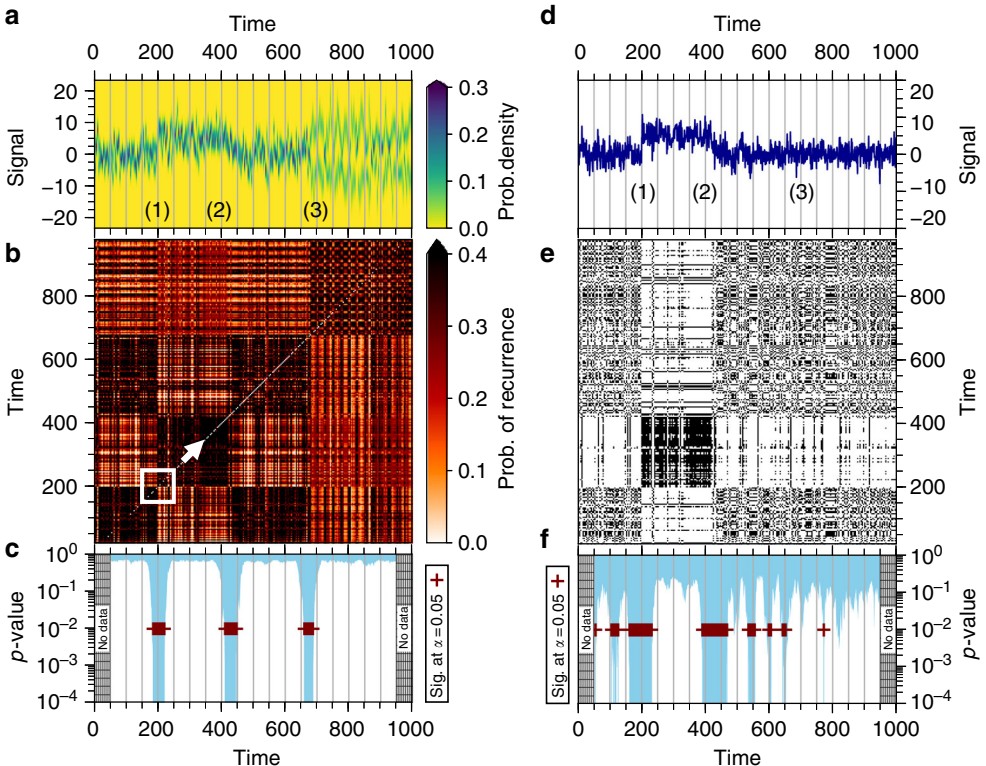

**Fig. 2** Detecting abrupt transitions with communities in networks of recurrence probabilities. The time series of PDFs $\{\rho_t^X\}_{t=1}^N$ for a synthetically generated noisy sinusoid (color map in (**a**)) and its mean (**d**). Three transitions are imposed: (1) a sudden jump at $T = 200$, (2) a linear decrease between $T = 400$ and $T = 450$, and (3) a change in the PDF to bimodality at $T = 675$. The probability of recurrence matrix $\hat{\mathbf{A}}$ in (**b**), estimated from the densities in (**a**), shows the modular structure resulting from the imposed transitions. The recurrence matrix $\mathbf{R}$ estimated from the mean time series (**e**) captures only the first two transitions. We detect the timing of the transitions by moving a sliding window (white box in (**b**)) of 100 time points and estimating the p-value (**c**, **f**) for a two-community structure under the null hypothesis of a random network. Statistically significant p-values (plus signs in (**a**)) are determined at a level $\alpha = 0.05$, and after accounting for multiple comparisons using Holm's method with the Dunn–Šidák correction factor (see Methods). In (**f**), the third transition is not detected and the first two are much more coarsely dated than in (**c**)

**Transitions in the equatorial central Pacific**. Our second real-world example involves recent climate data: the Niño 3.4 index is a standard index for estimating SST anomalies in the central-equatorial Pacific, calculated as the spatial averages of monthly gridded SST values in that region. Five consecutive 3-month (i.e., temporal) running averages of the index found above (below) a threshold of +0.5 K (−0.5 K) indicate El Niño (La Niña) conditions (Fig. 4b, d), two distinct phases of the ENSO which impact the climate worldwide. The transitions identified by our analysis (Fig. 4) show a relatively active period prior to 1906, after which their frequency decreases, indicating a complex interdecadal variability of the transitions themselves, most likely modulated by the PDO[18]. Based on a statistical coincidence analysis between the detected transitions and periods of 'phase locking' between the PDO and the ENSO (Methods, Supplementary Fig. 8), we reveal that the detected transitions are coincident with periods when the PDO and the ENSO were phase locked (green markers in Fig. 4b, d). This implicates the similarity of phases of the PDO and the ENSO as a potential factor that modulates ENSO dynamics, in addition to the phase of the PDO itself, which has been reported earlier[19] to increase the propensity for El Niño (La Niña) events when the PDO is in its positive (negative) phase.

**Transitions in the Asian summer monsoon in the Holocene**. The representation of observables as a sequence of PDFs is particularly valuable in paleoclimate time series analysis because of the inherent chronological uncertainties that hamper the

determination of the timings of short-lived events[20]. Here, we provide, for the first time, a transparent 'uncertainty-aware' framework to detect abrupt decadal-scale transitions in paleoclimate proxy records while considering dating uncertainties. We compare the transition detection results from speleothem datasets from the Dongge and Tianmen caves in China, and the Qunf Cave in Oman with the timings of well-known climatic events[21–23]. We detect significant shifts scattered through the Holocene (Fig. 5a–c) which likely correspond to weak ASM events reported in an earlier study[24] (blue squares in Fig. 5d). The weak ASM events have been postulated to be synchronous with ice-rafting episodes in the North Atlantic, known as Bond events (BEs, green squares in Fig. 5d)[24]. Our analysis confirms this hypothesis, allowing for the fact that the timings of the BEs themselves are still relatively poorly determined[23]. The BE at 1400 yrs BP potentially has a corresponding event in the Dongge cave record. However, this is not statistically significant when accounting for multiple comparisons. Although all the BEs have a potential corresponding event in the ASM records, the opposite is not true. We detect events of weakened ASM (ca. 6400–6800 yrs BP) which do not have a corresponding BE, suggesting additional influencing factors on ASM strength.

Our results further indicate a nontrivial spatial pattern in the hemispherical propagation of the events: the event at 8200 yrs BP, for example, is experienced first at Qunf, followed by Dongge, and then at Tianmen. We note, from Fig. 5, that the weak ASM event at 4200 yrs BP, well known as the '4.2k event'[25] is not detected at Qunf, primarily because the Qunf cave $\{\rho_t^X\}_{t=1}^N$ time series has

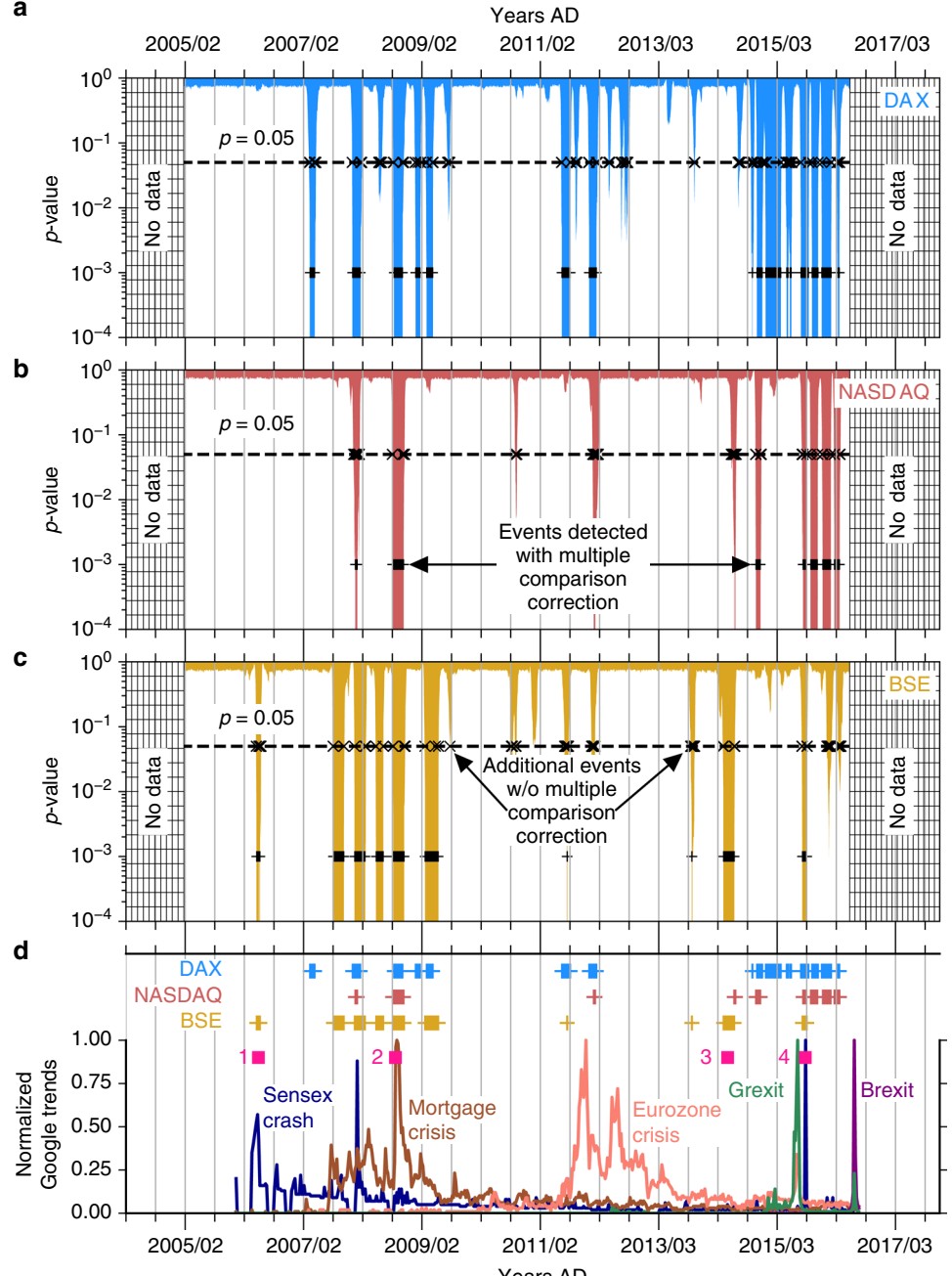

**Fig. 3** Abrupt transitions in financial stock indices. Applying our approach to three stock index datasets **a** DAX, **b** NASDAQ-100, and **c** S&P BSE SENSEX, we identify three major epochs centered around the 'mortgage crisis', 'Eurozone crisis', and the 'Grexit'/'Brexit' crises, as seen from the normalized Google trends data in (**d**). In each epoch we see a high number of statistically significant dynamical shifts at $\alpha = 0.05$ and these periods are interspersed with quiescent periods with far fewer of such shifts. Additional pink squares in (**d**) correspond to: (1) BSE SENSEX crash of 22 May 2006, (2) bankruptcy claim by Lehman Brothers on 15 September 2009, (3) Indian parliamentary elections from 7 April to 12 May 2014, and (4) SENSEX crash (1600 points) of 24 August 2015. The horizontal dashed lines in (**a**–**c**) indicate the confidence level $\alpha = 0.05$ of the statistical test. However, when multiple comparisons are taken into account (see Methods), only a subset of $p$-values below 0.05 are found to be significant (shown here with plus signs)

large dating uncertainties in the period between 3000 and 5000 yrs BP (Supplementary Note 4). These uncertainties result in large probabilities of recurrence for all pairs of time points within this period, such that there are no well-defined community structures indicative of abrupt transitions (Supplementary Fig. 9). This is not to say that the Qunf cave did not experience the 4.2k event, but rather to highlight the fact that: given the time series uncertainties, it is not possible to claim with statistical confidence whether or not there was an ASM event at the Qunf cave around 3000–5000 yrs BP. We can thus unambiguously link the detection

of an abrupt ASM transition to the time series uncertainties of the speleothem proxy record.

## Discussion

To summarize, we put forward a new approach to detect abrupt transitions in time series with uncertainties. Our approach is based on a novel representation of time series measurements as a sequence of PDFs $\{\rho_t^X\}_{t=1}^N$, which marks a shift from knowing the value of an observable at a given time to knowing how likely

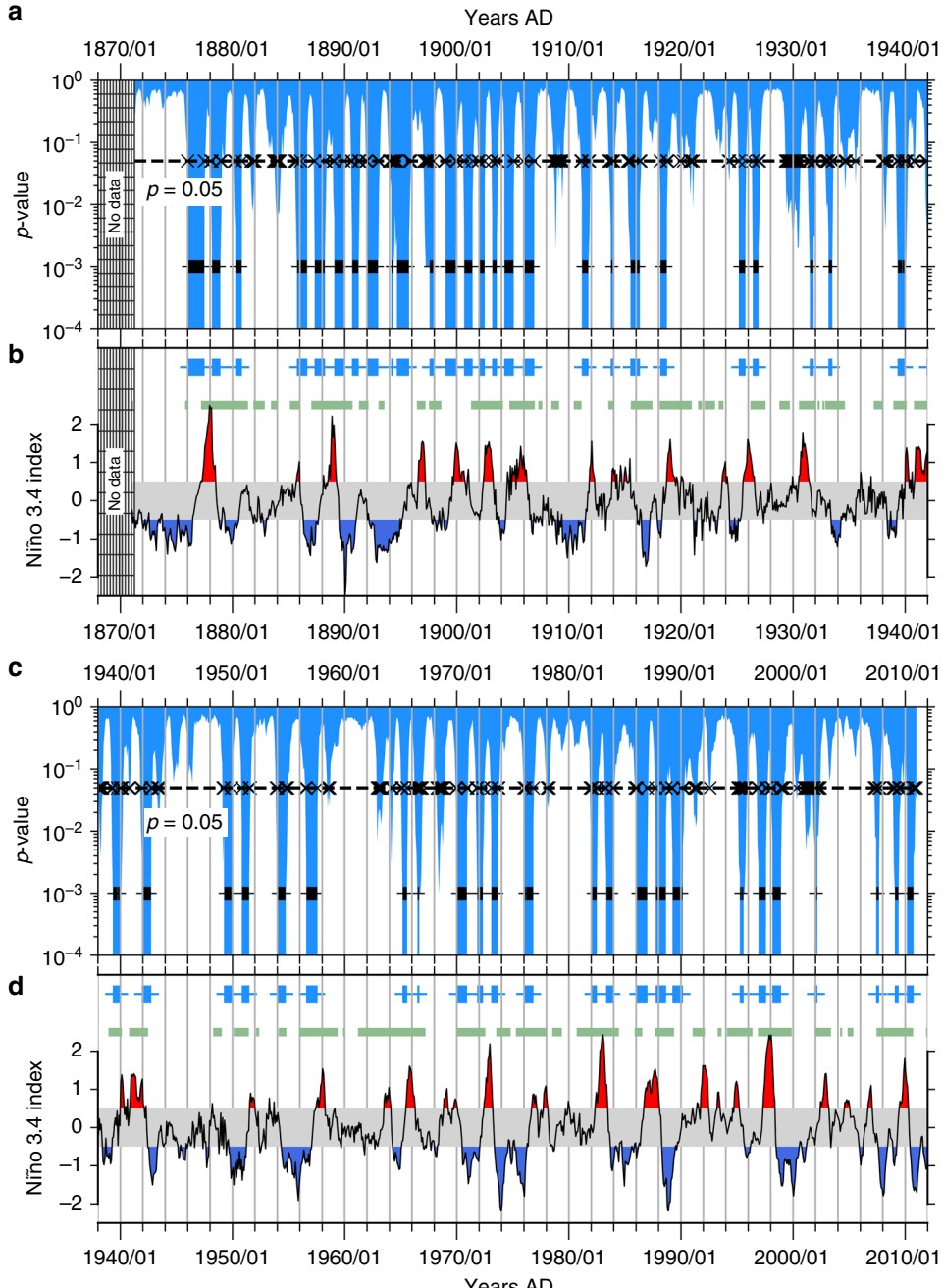

**Fig. 4** Abrupt transitions in the equatorial central Pacific. In (**a**, **c**), the plus and cross signs, and the horizontal dashed lines denote the same as in Fig. 3. We detect most of the transitions between El Niño (La Niña) phases, shown here as red (blue) shaded regions in (**a**, **c**) during the past 150 years: **a**, **b** From 1870 up to 1940. **c**, **d** From around 1940 to 2012. From around 1906, the transitions show an intermittent burst-like behavior, indicating a complex interdecadal variability of the transitions themselves. A statistical coincidence analysis reveals that the timings of the detected transitions are significantly coincident to the timings of phase-locked periods (shown here as green markers in (**b**, **d**)) between the PDO and the ENSO. This reveals a further potential aspect of the modulation of the ENSO by the PDO

was the observable to have a chosen value at that time. The PDF representation helps to assess the impact of uncertainties in time series estimates. We used the proposed PDF representation to tackle the question of identifying abrupt transitions mainly by utilizing the framework of recurrence analysis and estimating a network of probabilities of recurrence from the PDF sequence. Community structures in the estimated recurrence networks were then used as indicators of transitions in the time series.

We apply our approach to identify sudden shifts in three real-world examples chosen from diverse disciplines, viz., finance,

climatology, and paleoclimatology. The examples from finance served as a proof of concept of our approach, where we detected transitions in the stock exchanges from Frankfurt, New York, and Mumbai corresponding to well-known periods of political and economic crises. Applying our approach next to SST anomalies from the Niño 3.4 spatial region in the equatorial Pacific, we were able to detect abrupt transitions that were then found to be coincident with periods of phase locking between the Niño 3.4 and PDO indices. This reveals an additional aspect to the modulation of the ENSO by the PDO. Finally, we applied our method

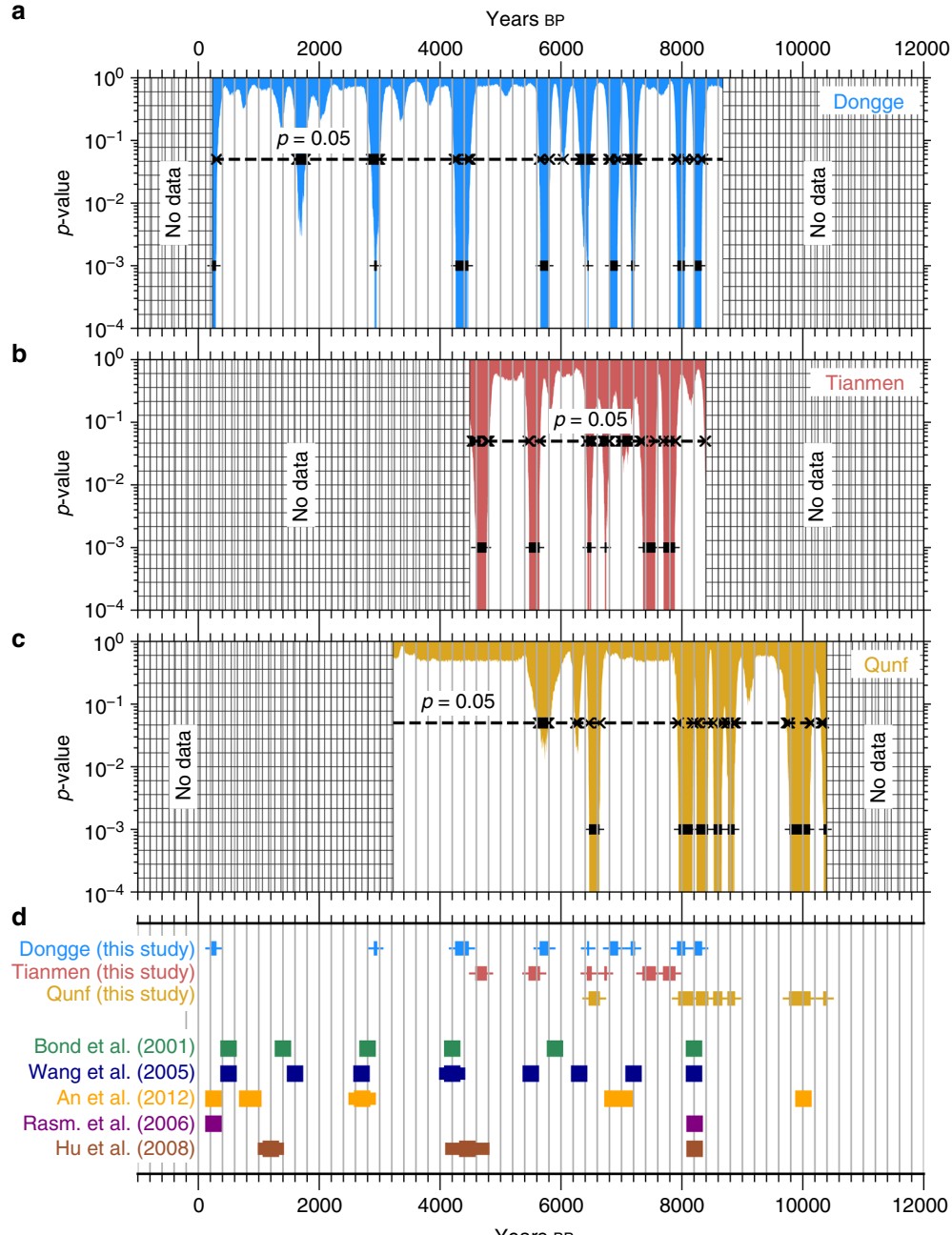

**Fig. 5** Abrupt transitions in paleoclimatic datasets. We apply our method to three paleoclimatic $\delta^{18}O$ proxy records from **a** Dongge, **b** Tianmen, and **c** Qunf caves in Asia. Statistically significant events (plus signs) show a scatter of events throughout the Holocene, corresponding to periods of weakened ASM (blue squares in (**d**)). The weak ASM events are postulated to be synchronous with BEs in the North Atlantic (green squares in (**d**)), a hypothesis that we are able to confirm with the results from our transition detection analysis. Barring the BE at 1400 yrs BP, all other BEs have a corresponding weak ASM that has been detected. At 1400 yrs BP, the Dongge cave record shows a potential dip in the p-value, but which is not statistically significant after accounting for multiple comparisons. Note that in (**a**–**c**), the plus and cross signs, and the horizontal dashed lines denote the same as in Fig. 3

to paleoclimatic proxies of the ASM and were able to validate for the first time, using an 'uncertainty-aware' framework, the hypothesis that sudden transitions in the ASM paleo-records are synchronous with the ice-rafting BEs originating in the North Atlantic.

The importance of considering the type and magnitude of the uncertainties of an observable in time series analyses cannot be stressed enough. By virtue of working with probability densities, we assert that our approach is innately geared to deal with uncertainties in measured data. This claim is well illustrated in the example of the Qunf cave record where the well-known '4.2k

event' was not detected, primarily due to the uncertainties in the initial $\{\rho_t^X\}_{t=1}^N$ sequence around that time period. By determining whether or not events 'show up' in a transition detection scheme, these uncertainties crucially influence subsequent inferences regarding the spatio-temporal propagation patterns of Holocene cold events—an as yet unanswered question in Holocene paleo-climatology. Our analysis takes a first step towards solving this issue, by showing how to incorporate uncertainties in detecting abrupt ASM transitions, and by helping us to clearly understand the importance of doing so. With our framework based on $\{\rho_t^X\}_{t=1}^N$, we hope to motivate a re-thinking of time series

approaches in terms of probabilities rather than point-like estimates.

## Methods

**Datasets**. We construct the synthetic dataset by considering 1000 independent 'nodes' of dynamics where, at each node, we have a deterministic sinusoid component and a normally distributed stochastic component,

$$x_0^u(t) = \sin(2\pi t/50) + 3\xi^u(t). \qquad (1)$$

Here $u = 1, 2, \ldots, 1000$ represents the node index, $t = 1, 2, \ldots, 1000$ denotes time, and $\xi^u(t) \sim \mathcal{N}(0, 1)$ is Gaussian white noise. This setup can be intuitively understood as being analogous to spatially gridded data from a region of interest, where the value of the observable at each grid location can be modeled as being offset from a local mean state (which is the sinusoid in Eq. 1 in our case) by some dynamic noise ($\xi^u(t)$ in Eq. 1). Next, at each location $u$, we impose three transitions, of which the first two, a sharp transition at $T = 200$ and a linear change between $T = 400$ and $T = 450$, change the baseline value of the sinusoid equally for all nodes. The third transition at $T = 675$, however, changes the baseline differently for different locations. Half of the nodes have their baseline raised and the other half have their baseline lowered. Formally, we can write the transitions as

$$\tilde{x}_0^u(t) = \begin{cases} x_0^u(t), & \forall u, 1 \le t \le 200 \\ x_0^u(t) + 5, & \forall u, 201 \le t \le 400 \\ x_0^u(t) + 45 - 0.1t, & \forall u, 401 \le t \le 450 \\ x_0^u(t), & \forall u, 451 \le t \le 675 \\ \begin{cases} 10 x_0^u(t), & 1 \le u \le 500 \\ -10 x_0^u(t), & 501 \le u \le 1000 \end{cases} & 676 \le t \le 1000 \end{cases} \qquad (2)$$

where $\tilde{x}_0^u(t)$ is the transition-imposed time series at location $u$. Finally, in order to simulate noisy measurements, we take $\tilde{x}_0^u(t)$ as the mean of a normal distribution with standard deviation equal to the measurement error 1.5, such that our final sampled time series $x_s^u(t)$ is: $x_s^u(t) \sim \mathcal{N}(\tilde{x}_0^u(t), 1.5)$. The 1000 time series of length 1000 time points each, obtained in this way, are then used to construct the PDF sequence in the next step.

The daily stock index data for the DAX (code: DAX, 30 companies from the Frankfurt Stock Exchange), NASDAQ-100 (code: NDX, 100 companies listed on the NASDAQ), and S&P BSE SENSEX (code: BSESN, 30 companies from the Bombay Stock Exchange) stock indices are obtained from http://finance.yahoo.com/ with their appropriate codes under the section 'Historical Prices'. Google trends data (Fig. 3d) were obtained from https://www.google.com/trends/ for the search queries: 'mortgage crisis', 'Eurozone crisis', 'Brexit', and 'Grexit' on 2 June 2016. The data were then normalized using a min–max transform such that they fall in the interval [0,1].

The monthly SST anomalies were obtained from the gridded SST data product Merged Hadley-NOAA/OI Sea Surface Temperature & Sea-Ice Concentration released by National Centers for Environmental Prediction[26], and available for free download at: http://www.esrl.noaa.gov/psd/data/gridded/data.noaa.oisst.v2.html. The Niño 3.4 region was extracted from the global data as those grid points falling within 5°N–5°S and 120°W–170°W. Anomalies were calculated with reference to the average climatology of the period: 1 January 1950 to 31 December 1979. The Niño 3.4 index data (Fig. 4b, d) was obtained from http://www.cpc.ncep.noaa.gov/data/indices/. The NCEI PDO index series (Supplementary Figs. 5 and 7) was obtained from https://www.ncdc.noaa.gov/teleconnections/pdo/.

The paleoclimate datasets for the Dongge[24], Tianmen[27], and Qunf[28] caves were obtained from the NOAA National Centers for Environmental Information (formerly the National Climatic Data Center). They are available for free download at: https://www.ncdc.noaa.gov/data-access/paleoclimatology-data.

**Constructing $\{\rho_t^X\}_{t=1}^N$ from measurements**. In the case of the synthetic example, for a given time slice $T = t$, we consider the values $\{x_s^u(t) \mid u = 1, 2, \ldots, 1000\}$ as the given set of measurements for that time instant and then obtain a kernel density estimate (using the Python scientific computing package SciPy) which yields an estimate of the PDFs $\rho_t^X$ for that time instant $t$.

For the financial datasets, we use the reported intra-day high $x_{\mathrm{hi}}(t)$ and intra-day low $x_{\mathrm{lo}}(t)$ values of the stock indices on a given day, and we postulate that, without any further information about the intra-day variations of the stock indices, the observed stock index values are noisy measurements of the underlying correct value of the index, which we assume to lie with equal probability anywhere inside the interval between $x_{\mathrm{lo}}(t)$ and $x_{\mathrm{hi}}(t)$. This results in the PDF sequence $\rho_t^X(x) = (x_{\mathrm{hi}} - x_{\mathrm{lo}})^{-1}$ for $x \in [x_{\mathrm{lo}}, x_{\mathrm{hi}}]$, and 0 otherwise.

In the SST dataset, for a given month in the SST data for the Niño 3.4 region, we take the spatially distributed SST anomaly values for that month and apply a kernel density estimation using an optimal bandwidth for Gaussian kernels with the Python toolkit Scikit-learn[29]. This results in an empirically estimated $\{\rho_t^X\}_{t=1}^N$ sequence constructed from the spatial distribution of SST values in a given month.

In the paleoclimate datasets, using the obtained proxy-depth and age-depth data, we estimate the posterior probability of the paleoclimatic proxy at a chosen time instant of the past using a Bayesian approach reported in an earlier study[30]. In

this approach, we consider the proxy, radiometric age, calendar age, and depth as the random variables $X$, $R$, $T$, and $Z$, respectively. In these terms, our quantity of interest is the probability $\rho(x \mid t)$, which for a speleothem dated with U/Th radiometric dates can be shown to be approximated by the Riemann sum

$$\rho(x \mid t) \approx \frac{\sum_{j=1}^{M} b_j\, w_t(z_j^x)\, \rho(x \mid z_j^x)}{\sum_{j=1}^{M} b_j\, w_t(z_j^x)} \qquad (3)$$

where $z_j^x$, $j = 1, 2, \ldots, M$ denotes the $M$ depth values at which the proxy measurements are made, and where $b_j$ is the width of the depth interval corresponding to $z_j^x$:

$$b_j = \frac{1}{2} \begin{cases} z_2^x - z_1^x & j = 1 \\ z_{j+1}^x - z_{j-1}^x & 1 < j < M \\ z_M^x - z_{M-1}^x & j = M \end{cases}. \qquad (4)$$

The weight term $w_t(z_j^x)$ in Eq. 3 encodes our beliefs as to how likely a chosen depth $z_j^x$ is given the calendar age $T = t$, i.e., $w_t(z_j^x) = \int \rho(r \mid t)\, \rho(r \mid z_j^x)\, \mathrm{d}r$. Thus, the probability that the proxy $X = x$ at a chosen time $T = t$ is expressed in terms of estimable or measured quantities. For the application to the paleoclimate datasets in this study, we take a regular time grid at 5-year intervals starting (ending) at the minimum (maximum) age measurement. We refer to the paper by Goswami et al. [30] for a more detailed discussion. For the current analysis, we use $\rho_t^X(x) := \rho(x \mid t)$ from Eq. 3.

**Network of recurrence probabilities**. We use the framework of recurrence analysis to analyze the chosen datasets. Typically, this is based on the construction of a recurrence matrix $\mathbf{R}$ whose elements $\mathbf{R}_{ij}$ are either 1 if the state $\mathcal{X}$ recurred (within an $\varepsilon$-neighborhood) at times $i$ and $j$, or 0 otherwise[11]. The recurrence matrix $\mathbf{R}$ can be used to classify and investigate various classes of complex dynamics. More recently, $\mathbf{R}$ has been shown to be interpretable as the adjacency matrix $\mathbf{A} = \mathbf{R} - \mathbf{1}$ of a complex network, called the recurrence network (RN), where the nodes are the time points of the observations and edges are placed between those pairs of time points which recur within an $\varepsilon$-neighborhood[12]. Here, $\mathbf{1}$ is the identity matrix of the same size as $\mathbf{R}$, which is subtracted from $\mathbf{R}$ to give an adjacency matrix $\mathbf{A}$ without self-loops.

In our case, since we consider time series with uncertainties, it is not possible to give a precise answer to the question whether time points $i$ and $j$ recurred, in the sense that we cannot answer this question with a 1 or a 0 as in a traditional recurrence analysis. We estimate instead the probability that $i$ and $j$ recurred in a chosen $\varepsilon$-neighborhood. A further point of difference with traditional recurrence analysis is that, till date, there does not exist any meaningful way to 'embed' a time series of probability densities, and we thus estimate the recurrence probabilities in the following without embedding our data. Therefore, from here on, we use the simple scalar difference between the observable at times $i$ and $j$ as our distance metric in order to define a 'recurrence'. For the sake of clarity, we define as $X_i$ and $X_j$ the random variables describing the observable at time points $i$ and $j$ respectively, and $Z_{ij} := X_i - X_j$ as the random variable describing the difference of the observable at $i$ and $j$. The probability of recurrence $Q_{ij}(\varepsilon) := Prob(|Z_{ij}| \le \varepsilon)$ (where $|\cdot|$ denotes the absolute value) is

$$Q_{ij}(\varepsilon) = \int_{-\infty}^{+\infty} \rho_i^X(x_i) \int_{x_i-\varepsilon}^{x_i+\varepsilon} \rho_{j|i}^X(x_j \mid x_i)\, \mathrm{d}x_i\, \mathrm{d}x_j = \int_{-\infty}^{+\infty} \int_{x_i-\varepsilon}^{x_i+\varepsilon} \rho_{ij}^X(x_i, x_j)\, \mathrm{d}x_i\, \mathrm{d}x_j, \qquad (5)$$

where the joint probability density $\rho_{ij}^X(x_i, x_j)$ is not provided and is also not uniquely estimable from the marginal densities $\rho_i^X(x_i)$ and $\rho_j^X(x_j)$ alone. To overcome this limitation we use the results from an earlier study by Williamson and Down [31] who show how upper and lower bounds on the cumulative distribution function (CDF) of $X_i - X_j$ can be derived using only the marginal CDFs for $X_i$ and $X_j$, denoted here as $P_i^X$ and $P_j^X$ respectively. In the following, we thus first describe the probability of recurrence $Q_{ij}(\varepsilon)$ in terms of the CDF of $Z_{ij} = X_i - X_j$ and then use the bounds given by Williamson and Down [31] to derive precise bounds for $Q_{ij}(\varepsilon)$ itself.

The probability of recurrence $Q_{ij}(\varepsilon)$ can be seen as the total probability that $Z_{ij} = X_i - X_j$ falls in the interval $[-\varepsilon, \varepsilon]$,

$$Q_{ij}(\varepsilon) = Prob(|Z_{ij}| \le \varepsilon) = Prob(Z_{ij} \le \varepsilon) - Prob(Z_{ij} \le -\varepsilon), \qquad (6)$$

which implies

$$Q_{ij}(\varepsilon) = P_{ij}^Z(\varepsilon) - P_{ij}^Z(-\varepsilon), \qquad (7)$$

where $P_{ij}^Z(z_{ij}) := Prob(Z_{ij} \le z_{ij})$ is the CDF of $Z_{ij}$, and $z_{ij} \in (-\infty, \infty)$.

The upper bound $M_{ij}$ and the lower bound $m_{ij}$ for the CDF $P_{ij}^Z(z_{ij}) \forall z_{ij}$ is obtained using the results from Williamson and Down [31] as,

$$M_{ij}(z_{ij}) = \min\left\{\inf_v f_{ij}(v, z_{ij}), 0\right\} + 1, \text{ and} \tag{8}$$

$$m_{ij}(z_{ij}) = \max\left\{\sup_v f_{ij}(v, z_{ij}), 0\right\}, \tag{9}$$

where $f_{ij}(v, z_{ij}) = P_i^x(v) - P_j^x(v - z_{ij})$. These bounds ensure that, for all values of $Z_{ij} = z_{ij}$, $P_{ij}^Z(z_{ij}) \in [m_{ij}(z_{ij}), M_{ij}(z_{ij})] \subseteq [0, 1]$. Using these bounds and combining them with Eq. 7, we find that the upper bound $q_{ij}^u(\varepsilon)$ and the lower bound $q_{ij}^l(\varepsilon)$ for the probability of recurrence $Q_{ij}(\varepsilon)$ can be written as,

$$q_{ij}^u(\varepsilon) = \min\{M_{ij}(\varepsilon) - m_{ij}(-\varepsilon), 1\}, \text{ and} \tag{10}$$

$$q_{ij}^l(\varepsilon) = \max\{m_{ij}(\varepsilon) - M_{ij}(-\varepsilon), 0\}. \tag{11}$$

Given a recurrence threshold $\varepsilon$, the bounds $q_{ij}^u(\varepsilon)$ and $q_{ij}^l(\varepsilon)$ constrain the probability of recurrence $Q_{ij}(\varepsilon)$ such that $Q_{ij}(\varepsilon) \in [q_{ij}^l(\varepsilon), q_{ij}^u(\varepsilon)] \subseteq [0, 1]$. We drop $\varepsilon$ in the following for notational clarity and with the understanding that $\varepsilon$ is fixed.

First, we assume the probability $Q_{ij}$ itself to be a random variable distributed in the obtained interval $[q_{ij}^l, q_{ij}^u]$, in a way unknown to us, and define the PDF within these bounds as $\rho_{ij}^Q(q_{ij})$. Next, assuming $\mathbf{A}$ to be the true but not estimable adjacency matrix of the system's recurrence network, we write down the probability of having a link between $i$ and $j$ as

$$Prob(\mathbf{A}_{ij} = 1) = \int_{q_{ij}^l}^{q_{ij}^u} \rho_{ij}^{\mathbf{A}|Q}(\mathbf{A}_{ij} = 1 q_{ij}) \, \rho_{ij}^Q(q_{ij}) \, dq_{ij} = \int_{q_{ij}^l}^{q_{ij}^u} q_{ij} \, \rho_{ij}^Q(q_{ij}) \, dq_{ij} = \mathbf{E}_{\rho_{ij}^Q}[Q_{ij}], \tag{12}$$

where we use the notation $\rho_{ij}^{\mathbf{A}|Q}(A_{ij} = 1 \,|\, q_{ij})$ to denote the probability that $\mathbf{A}_{ij} = 1$ given $Q_{ij} = q_{ij}$ and use the result that $\rho_{ij}^{\mathbf{A}|Q}(A_{ij} = 1 \,|\, q_{ij}) = q_{ij}$. The final result of Eq. 12 shows how the total probability that $\mathbf{A}_{ij}$ equals 1 is simply the expectation of $Q_{ij}$ evaluated with respect to $\rho_{ij}^Q(q_{ij})$.

Assuming that $Q_{ij}$ is distributed symmetrically around the mean in the interval $[q_{ij}^l, q_{ij}^u]$, the total probability that the observable at $i$ and $j$ (from Eq. 12) is

$$Prob(\mathbf{A}_{ij} = 1) = \mathbf{E}_{\rho_{ij}^Q}[Q_{ij}] = \frac{q_{ij}^l + q_{ij}^u}{2}, \tag{13}$$

which allows us to define an estimator $\hat{\mathbf{A}}$ of the probabilities of recurrence of the observable $X$ and interpret it as the adjacency matrix of a network whose nodes are the time points of observation and whose edge weights are defined as

$$\hat{\mathbf{A}}_{ij}(\varepsilon) := \begin{pmatrix} \frac{1}{2}\left(q_{ij}^l(\varepsilon) + q_{ij}^u(\varepsilon)\right) & i \neq j, \\ 0 & i = j \end{pmatrix}. \tag{14}$$

Here, we put $\hat{\mathbf{A}}_{ii} = 0$ to avoid self-loops in the network. The elements of $\hat{\mathbf{A}}_{ij}$ encode the total probability that time points $i$ and $j$ have recurred within an $\varepsilon$-neighborhood, taking into account the uncertainties in the dataset.

In order to estimate the networks of recurrence probabilities for the applications, we use a bisection routine to arrive at a suitable $\varepsilon$ threshold which results in a pre-specified link density of the RN. The link densities chosen are: (1) synthetic example, 30%, (2) financial datasets, 24%, (3) SST dataset, 25%, and (4) paleoclimatic datasets, 30%.

**Detecting abrupt transitions using recurrence network community structure**. Block-like structures in recurrence plots have been suggested to encode the occurrence of an abrupt transition in the dynamics of the observable under consideration[32]. In the RN framework, such structures correspond to communities, defined in the sense of Newman[17] as those parts of a network which have a higher link density within themselves than to the rest of the network. RN communities represent a time period in which the states of the system are closer to each other than to the rest and thus, they correspond to stable regimes of dynamics and their borders are the time points at which the system transited between regimes. However, as our interest is confined solely to the question of the existence of an abrupt transition at the midpoint of the time period under consideration, we do not need to do a typical 'community detection' by taking into account all possible partitions of the network. Rather, for our purposes it suffices to move a sliding window over the dataset (Fig. 2b), and after extracting the portion of $\hat{\mathbf{A}}$ which falls within that window, to test whether the two halves of the network (before and after the midpoint) form two communities that are unlikely to have occurred due to chance. We use the within-community link fraction $S$ as an indicator of community structure. We expect those windows with a high value of $S$ to have very low $p$-values. A low $p$-value, obtained at the end of statistical hypothesis testing for the window under consideration, would imply that we fail to accept the null hypothesis that the two-community structure observed in that window occurred by chance.

We consider the sliding window at a position $k$, where it covers the time period $T \in [t_1, t_2]$, and define the midpoint of the window as

$$t_m = \left\lfloor \frac{t_1 + t_2}{2} \right\rfloor, \tag{15}$$

where $\lfloor \cdot \rfloor$ denotes the greatest integer function, which gives the largest integer less than or equal to the given function argument. Defining $\hat{\mathbf{A}}^k$ as the portion of the full adjacency matrix $\hat{\mathbf{A}}$ which covers the interval $[t_1, t_2]$, we can estimate the within-community link fraction $s^k$ for the window at $k$ as

$$s^k = \frac{\sum_{i,j=t_1}^{t_m} \hat{\mathbf{A}}_{ij}^k + \sum_{i,j=t_m}^{t_2} \hat{\mathbf{A}}_{ij}^k}{\sum_{i,j=t_1}^{t_2} \hat{\mathbf{A}}_{ij}^k}. \tag{16}$$

The extent to which the value of $S$ obtained from the data in a particular window $(:= s_{obs}^k)$ is governed by a dynamical transition and not by randomness is quantified using the $p$-value $(:= p_{s_{obs}^k})$ of a statistical test with the null hypothesis $H_0$: $S$ is determined by the degree sequence of the given network. Here, the term degree sequence denotes the sequence of the total weight of links for each node. To simulate this null hypothesis, we use the Python Igraph module which allows us to generate random networks with a prescribed degree sequence. In each window $k$, we generate 1000 such random surrogate networks and obtain an ensemble $s_{sur}^k$ of values of $S$, from which we can estimate $p_{s_{obs}^k}$ using the percentile of $s_{obs}^k$ in the distribution $S_{sur}^k$,

$$p_{s_{obs}^k} = 1 - \frac{F_{sur}^{S^k}(s_{obs}^k)}{100}, \tag{17}$$

where $F_{sur}^{S^k}(\cdot)$ is a percentile function, which returns the percentile of the given argument within the distribution of the 1000 $s_{sur}^k$ values generated in the previous step. Here, we assume that 1000 is a sufficient number of random realizations for the following approximation to hold:

$$P_{null}^{S^k}(s^k) \approx \frac{F_{sur}^{S^k}(s^k)}{100}, \tag{18}$$

where $P_{null}^{S^k}$ is the CDF of the within-community link fraction $S^k$ corresponding to the null hypothesis. Finally, to estimate the statistically significant windows for a chosen confidence level $\alpha$, we apply the Holm's method to correct for multiple comparisons[33] along with the Dunn–Šidák correction factor[34] (Figs. 2, 3 and 4).

The sizes of the sliding window were different for the different applications: synthetic example, 100 time points; financial datasets, 60 time points (approximately 2 months); SST data, 30 time points (2.5 years); and paleoclimatic datasets, 100 time points (500 years). The confidence level alpha was set at $\alpha = 0.05$ for all tests.

**Coincidence analysis of the ENSO transitions**. Here, we define phases for the Niño 3.4 index and the PDO index (Methods: Datasets) in the sense of Maraun and Kurths[35]. In this approach, a low-pass forward–backward Butterworth filter is applied on the index time series $\{y_t\}_{t=1}^N$ such that all frequencies higher than $\frac{1}{12}$ months$^{-1}$ are removed. The filtered time series $\tilde{y}_t$ is then used to obtain the instantaneous phase $\phi_t$ (Supplementary Fig. 7),

$$\phi_t = \arctan\left(\frac{H(\tilde{y}_t)}{\tilde{y}_t}\right) \tag{19}$$

where $H(\cdot)$ denotes the Hilbert transform. The time derivative $\frac{d(\Delta\phi_t)}{dt}$ of the instantaneous phase difference $\Delta\phi_t = \phi_t^{PDO} - \phi_t^{ENSO}$ helps to identify periods of 'phase locking' where $\Delta\phi_t \approx 0$ (Supplementary Fig. 8a). We consider the two index time series, representing ENSO and PDO respectively, to be phase locked when $\frac{d(\Delta\phi_t)}{dt}$ falls between the 25th and 75th percentile of all $\frac{d(\Delta\phi_t)}{dt}$ values (Supplementary Fig. 8b). Next, we test whether the phase-locked time periods are significantly coincident with the abrupt transitions detected using our current method on the PDF series of SST anomalies from the Niño 3.4 region.

We consider a detected transition to be coincident if it occurs within 31 days of a time point of phase locking. In total, 216 such coincidences are identified. To test whether this could occur simply by chance, we randomize the timings of the detected transitions 50,000 times and compute the number of coincidences each time. This results in a null distribution of coincidences occurring purely by chance (Supplementary Fig. 8c). At a significance level of 5%, we find that the observed number of coincidences is significantly higher than that possible by pure random chance, which validates the hypothesis that the detected abrupt transitions in the Niño 3.4 region are significantly coincident with periods of phase locking between the PDO and the ENSO.

**Code availability**. All numerical computation required to implement the ideas presented in this study have been coded in Python 2. All necessary codes are available on request from the corresponding author.

**Data availability**. We have used previously released, freely available datasets in our study. The various data sources have been listed in Methods, Datasets. In case of any difficulty in obtaining the datasets mentioned above, the corresponding author can provide the data used upon request.

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

## Acknowledgements

This paper was developed within the scope of the IRTG 1740/TRP 2011/50151–0, funded by the DFG/FAPESP, and the DFG project IUCLiD (Project No. DFG MA4759/8). N.B. acknowledges funding by the Alexander von Humboldt Foundation, the German Federal Ministry for Education and Research, and the DFG. J.K. acknowledges financial support from the Government of the Russian Federation (Agreement No. 14.Z50.31.0033). S.F.M.B., N.M., and B.G. have received funding from the European Unions Horizon 2020 Research and Innovation programme under the Marie Skłodowska-Curie grant agreement no. 691037 (project QUEST). B.G. was partially supported by the MWFK Brandenburg.

## Author contributions

B.G. wrote the main text and prepared the figures. B.G., N.M., and J.H. developed the estimation procedure of the probabilities of recurrence. B.G., A.R., and N.B. developed the RN community structure approach to detect abrupt transitions, and the associated significance tests. S.F.M.B. preprocessed the paleoclimate datasets, constrained the Holocene events reported in the scientific literature, and drew conclusions from the paleoclimatic results. All authors discussed the results, drew conclusions, and contributed to the manuscript text. All authors were involved in deciding the context and narrative within which the main results have been communicated.

## Additional information

**Competing interests:** The authors declare no competing financial interests.

