## [Peer Review File · Nature Communications]

Reviewers' comments:

Reviewer #1 (Remarks to the Author):

I enjoyed reading this paper and support publication in Nature Communications. Prior to publication however, I would like the authors to address the following points (in no particular order):

- The sentence "More recently... "(lines 214-216) needs a reference to the first such statement that a recurrence (either epsilon or k-neighbour?) matrix can be treated as a network adjacency matrix
- The paragraph from 218-230 is difficult to interpret (as this is the main statement of the core of their method, this needs to be strengthened. In particular: (1) is $P_t(x)$ a cdf or is it cumulative pdf over a time window? (2) the "earlier study" is actually quite a separate piece of work and should be references somehow more clearly. (3) Is X_i the random variable of x_i ? (4) what is z_{ij} ?
- From reading this paragraph one is left with three possible interpretations of the main gist of this paper - pdfs are estimated from a moving window over the data and statistically tested for stationarity, or, somehow a weighted method is applied to obtain an instantaneous pdf, or, the carefully chosen metric for "similarity" in the recurrence matrix allows the recurrence matrix itself to be interpreted as a point estimate of said pdf. Which is not clear at this point.
- What *is* the choice of similarity metric? Is the data embedded?
- It would be useful to unravel (6) in terms of the input time series.
- the parameter epsilon is chosen to meet a proscribed link density - but how is that link density chosen (it seems to differ between applications and is, in every case, quite high)?
- how does this method compare to quadrant scans of Rapp and colleagues (see https://www.researchgate.net/publication/274905301_Hierarchical_Transition_Chronometries_in_the_Human_Central_Nervous_System)
- how are the network communities detected? why use this method? what effect does alternative choices have?
- Ditto - the choice of the sliding window?
- is the 100 time points in Figure 2 equal to 50 data?
- the paper should be carefully proof-read (the authors refer to it as a "letter" on page 2, and on page 7 use the awkward phrase "we have on our hands...")
- the scale of the colour encoding of the the sliding pdfs in the supplementary information is not particularly helpful - apart from the paleo climate proxies, they all look just like time series on a yellow background.

Reviewer #2 (Remarks to the Author):

Please see the attached report. My recommendation is that the paper be rejected but the authors invited to submit a new version after a major revision.

Report on NCOMMS-17-13021-T: *Abrupt transitions in
time series with uncertainties*
by Goswani et al.

In the paper the authors consider the problem of modeling time series when there are, using the language of the paper, abrupt transitions in the dynamics of the underlying process. The provided real-world examples of such time series include financial stock indices (DAX, Nasdaq-100 and BSE SENSEX are used), sea surface temperatures anomalies from the Niño 3.4 region and paleoclimatic proxies, all of which are described in detail in the paper.

The main contribution of the paper is the proposal of modeling time series by considering underlying probability densities rather than the observables themselves. The authors highlight the application of their method to paleoclimate time series analysis and claim that this is the first time a “uncertainty-aware” framework is applied in the setting of paleoclimate proxy records.

The proposed method for modeling a time series, supposedly taking into account what the authors refer to as time series uncertainties, is as follows: From an observed time series $\{x_t\}_{t=1}^n$, the authors model a corresponding sequence of probability densities $\{\rho_t(x)\}_{t=1}^n$; for each t , $\rho_t(x)$ is supposed to represent a probability density from which the observed x_t can be viewed as an outcome (more specifically, the density of a random variable for which x_t is viewed as an outcome). Moreover, abrupt transitions are detected by conducting a so-called recurrence analysis. The densities $\rho_t(x)$ are used to estimate the probability of recurrence (in an ϵ -neighborhood sense) for all time points, resulting in an estimated $n \times n$ -matrix $\hat{\mathbf{A}}$ of such probabilities. Interpreting this matrix as the adjacency matrix of a network, the authors then characterize abrupt transitions by the presence of so-called communities in this network; communities are characterized by having higher similarity within themselves than the rest of the time series, indicating time intervals with distinct dynamics.

The novelty of the paper is stated (both implicitly and explicitly) to be (i) the modeling of a sequence of probability densities based on an observed time series, (ii) the combination of (i) and a recurrence network analysis, and (iii) applications to paleoclimate time series. As my expertise covers mainly the modeling part this review will focus on that part of the paper.

Regarding the modeling step (i), the section “Constructing $\rho_t(x)$ from measurements” is at times confusing. There are inconsistencies, at least in the language used, in whether quantities are considered as outcomes of a stochastic process or as deterministic. For the daily stock index data, the use of a uniform distribution over the interval $[x_{lo}(t), x_{hi}(t)]$, taking into consideration that financial data typically exhibits heavy tails. For the paleoclimatic proxy records, in equation (9) the quantities $w_t(z_j^x)$ are not defined in the paper. In general, it is not obvious to me how much is added to the modeling by

considering densities rather than a model for the observed values, since you still have to make a choice for what type of density to use for a given time series (as opposed to the dependence structure or similar). Moreover, the authors should be more specific in addressing the aim of their method: Is it primarily for prediction, for inference etc.? This is never mentioned in the paper, although it seems as if it is mainly inference that is intended.

For the recurrence analysis, it is again quite difficult to follow the description of the method. Especially since the authors are not clear on what are random variables, constants, distributions etc (see comments at the end of this review). In the description of how to detect communities from the estimated adjacency matrix, it is not clear what the null hypothesis and involved random variables are. What does it mean to “specify the communities before we measure S ”? Moreover, the sampling from the degree-configuration model could be a computationally costly task for large samples (a large number of nodes). How is the sampling done, using a vanilla-type algorithm? How do the authors check that sampling has converged to something close to the true distribution?

Although the report may seem negative, I am interested in the authors’ ideas and the proposed methodology. However, for it to warrant publication in a journal like this it is my opinion that the mathematical level must be raised and the overall presentation improved. Alternatively, the authors could state early on that the modeling in the paleoclimate-setting is the main focus and put the emphasis on those results; for financial time series there are already ways of taking into account shifts in the underlying dynamic and a focus on such time series would then warrant a more in-depth comparison. As of now, I read the paper as primarily wanting to introduce a methodology, which is poorly described. The authors should also put more effort into comparing their proposed method to other ways of detecting changes in the dynamic of a time series (see the vast literature on change-point-detection) and clearly stating what the improvements are. Some (additional) relevant questions are: How does one choose what type of density to use in a real-world example? Are there any performance guarantees or diagnostics that can be used?

With the above motivation I recommend that the paper **be rejected but the authors invited to submit a new version of the paper after a major revision**. However, as I am less familiar with the mathematical standard of the journal, I leave it to the AE to decide whether this is indeed a relevant argument.

I end this review with some minor comments on the text.

- l. 163) x_0 a stochastic process and thus referring to it as a “mean” is not proper terminology.
- l. 167) x_0 has a stochastic process, hence it is a poor choice of words to refer to the $x_s(t)$ as “mean(s)” - this suggests a deterministic quantity (see comment on l. 163).
- l. 195) The notation “ $\rho_t(x) \sim \mathcal{N}(x_s(t), \sigma_s)$ ” typically signifies that the former is a random variable distributed according to the latter. Change to something like “ $\rho_t(x)$ is taken as the density of a $\mathcal{N}(x_s(t), \sigma_s)$ -distribution”.
- l. 224) $P_{Z_{ij}}$ is here referred to as a distribution but later (l. 226) as a scalar value: $P_{Z_{ij}} \in [m_{ij}, M_{ij}]$. Be clear on what you mean. For example, is the interval the support of the distribution?

- l. 231) Q_{ij} is a probability but here treated as a random variable. Are the authors really considering the conditional probabilities given the observed values of X ?
- Eq. (5)) Define also $\rho(\mathbf{A}_{ij} = 1|q_{ij})$. Readily apparent from the context but should still be explicitly defined (since ρ is used for other things as well).
- l. 236) Do you mean ‘uniformly’ here, rather than ‘symmetrically’? If not, how is the probability mentioned on l. 237 obtained?
- l. 258) The authors mentions the networks ‘degree-sequence’ without defining this quantity. Readers specialized in either time series or the applied areas addressed in the paper are not necessarily familiar with graph-theoretic concepts.

Response to reviewers' comments
NCOMMS-17-13021-T
Abrupt transitions in time series with uncertainties

Bedartha Goswami, Niklas Boers, Aljoscha Rheinwalt,
Norbert Marwan, Jobst Heitzig, Sebastian F. M. Breitenbach, and Jürgen Kurths

We would like to thank the two reviewers for their insightful comments on our manuscript. We have incorporated their suggestions and are convinced that, in the process, our manuscript has improved considerably. We go through the points raised by both reviewers in the sections below and provide our responses and references to changes/revisions in the text accordingly.

Reviewer #1

1. *I enjoyed reading this paper and support publication in Nature Communications. Prior to publication however, I would like the authors to address the following points (in no particular order):*

We thank the reviewer for this positive response. We address the points raised by the reviewer in the paragraphs below.

2. *The sentence “More recently. . .” (lines 214-216) needs a reference to the first such statement that a recurrence (either epsilon or k-neighbour?) matrix can be treated as a network adjacency matrix.*

We have added an appropriate reference to the study by Marwan et al. [1] which first put forward the idea of considering a recurrence matrix as the adjacency matrix of a network embedded in the phase space of the dynamics. We note, however, that this reference was already cited in the main text, in lines 78–79 of the earlier version, and is to be found in line 102 of the revised version.

Also, we use the ε -neighbourhood definition of recurrences all throughout our paper.

3. *The paragraph from 218-230 is difficult to interpret (as this is the main statement of the core of their method, this needs to be strengthened.)*

We agree with this observation and apologize for the ambiguous description. The previous version of our manuscript was originally submitted as a letter, and due to the space restrictions of that format we had to condense the Methods section considerably. As a result, we had to shorten the description of the network of probabilities of recurrence. In the revised version of our paper, we have now expanded the derivation of the bounds on recurrence probabilities appropriately in order to make it more comprehensible.

- (a) ***In particular: (1) is $P_t(x)$ a cdf or is it cumulative pdf over a time window?***

$P_t(x)$ is the cumulative distribution function (CDF) corresponding to the probability density function (PDF) $\varrho_t(x)$ at a chosen time instance t . However, in our current version we do not refer to this formal notation and we simply use the abbreviation CDF when necessary. In the case of the difference variable $Z_{ij} = X_i - X_j$ (cf. Equation 6 of the revised paper) we have now changed the notation of the CDF to P_{ij}^Z .

- (b) ***(2) the “earlier study” is actually quite a separate piece of work and should be references somehow more clearly.***

We now cite the earlier study by Williamson & Down [2] in more detail, taking care to explain the context in which we use the results derived in their paper.

- (c) ***(3) Is X_i the random variable of x_i ?***

X_i is the random variable describing the observable at a chosen time instance $T = i$, whereas x_i is a particular value of this variable. We use this notation with the understanding that it is conventional to use upper case letters to denote the (abstract) random variable and lowercase letters to denote particular values of that same random variable, e.g., the conditional probability of $A = a$ given that we have fixed/observed/chosen $B = b$ can be written as $P(A = a|B = b)$ which is equivalent to writing $P(a|b)$.

- (d) ***(4) what is z_{ij} ?***

z_{ij} is a particular instance of the random variable Z_{ij} (cf. our response 3(c)) which describes the difference between the observable X at time points i and j , i.e., $Z_{ij} = X_i - X_j$. We apologize for the confusing text in the earlier version. In our current version, all relevant variables have been defined properly before use.

4. ***From reading this paragraph one is left with three possible interpretations of the main gist of this paper - pdfs are estimated from a moving window over the data and statistically tested for stationarity, or, somehow a weighted method is applied to obtain an instantaneous pdf, or, the carefully chosen metric for “similarity” in the recurrence matrix allows the recurrence matrix itself to be interpreted as a point estimate of said pdf. Which is not clear at this point.***

There has been a serious misunderstanding with regard to the above mentioned paragraph (i.e., lines 218–230 of the earlier version of the paper). We do not estimate PDFs from a moving window. In the above mentioned paragraph we describe how to obtain the probability of recurrence given the marginal PDFs ϱ_i^X and ϱ_j^X at two time points i and j respectively. These two PDFs are instantaneous but are not obtained using any weighting scheme. Rather, they are ‘constructed’ in various ways depending on the specific example at hand (cf. subsection “Constructing $\{\varrho_t^X\}_{t=1}^N$ from measurements” in the Methods section). Moreover, the entries of the recurrence matrix $\hat{\mathbf{A}}_{ij}$ are not point estimates of the above mentioned PDFs. They denote the corresponding estimates of the probabilities of recurrence (using the method described in the subsection “Network of recurrence probabilities”) given the marginal PDFs ϱ_i^X and ϱ_j^X . The only instance where we use moving windows in our study is when we quantify the block structure of the recurrence networks in order to detect the timings of abrupt transitions.

We have modified the description of the method extensively with a strong focus on terminology and notation. We hope that the revised version allows for a clearer understanding of our approach.

5. *What ***is*** the choice of similarity metric? Is the data embedded?*

We do not use a ‘similarity’ metric but rather a distance metric in order to define a recurrence between time points i and j when they are ‘closer’ than a threshold ε . We simply use the scalar difference between the observable X at times i and j and use the notation $|X_i - X_j| \leq \varepsilon$ where $|\cdot|$ denotes the absolute value. We use the scalar difference as it is the most natural choice in our situation, since we do not embed the time series. This is because, till date, there is no meaningful way to embed the series of probability densities $\{\varrho_t^X\}_{t=1}^N$. E.g., Takens’ embedding theorem is defined for time series of point-like observables $\{x_t\}_{t=1}^N$ and not $\{\varrho_t^X\}_{t=1}^N$.

The above points are now highlighted in the subsection “Network of recurrence probabilities” of the revised version (lines 300–304).

6. *It would be useful to unravel (6) in terms of the input time series.*

[Note: Equation 6 of the earlier version is renumbered to Equation 14 in the revised version. We use the equation numbers of the revised version in this response document.]

We respectfully disagree with the reviewer on this point. We feel that substituting the various terms on the right-hand side of Equation 14 until we arrive at an expression with $\varrho_t^X(x)$ will result in an unwieldy and cumbersome equation. In our opinion, the system of equations from Equations 6–13 are already well-equipped (when taken together for interpretation) to describe our proposed approach.

7. *the parameter epsilon is chosen to meet a proscribed link density - but how is that link density chosen (it seems to differ between applications and is, in every case, quite high)?*

The choice of the link density is motivated by the timescales at which we wish to detect transitions. The transitions detected in our study are determined by: (i) the size of the sliding window (larger size of window would detect transitions that separate longer periods of similar dynamics), and (ii) the link density of the overall recurrence network. A network of higher link density will lead to a ‘denser’ recurrence matrix as compared to one with a lower link density. A pair of adjacent communities that were detectable at lower link densities might get ‘filled up’ to form a single community at a higher link density, which would cover up the transition that existed between the two at lower link densities.

Our choice of the link densities in all of the examples are thus determined by the timescales of communities that we wish to use in order to detect transitions. In the finance example we focus on monthly timescales, whereas in the ENSO example, we focus on interannual timescales, and in the paleoclimate example we focus on centennial timescales. Moreover, we would like to point out that in all of these cases, the results as well as all subsequent inferences are robust to small changes in the link density, i.e., our inferences are not affected by changing the link density in small amounts.

We agree with the reviewer that the link densities used in our study, ca. 25–30%, appear quite high when compared to typical link densities of ca. 10–15 % (equivalent to the recurrence rates) in recurrence-plot-based studies. This is because in a traditional binary

recurrence matrix, there is a high number of zeros whereas in the probability of recurrence matrices used in our study, there are typically a high number of entries that are close to zero but not quite zero, which then sum up to contribute to a comparatively higher link density while resolving more or less the same pattern as a traditional binary recurrence plot of a lower link density.

8. ***how does this method compare to quadrant scans of Rapp and colleagues (see¹)***

We thank the reviewer for pointing us to the very interesting study by Rapp et al. [3], which we have now included in our list of references (it is reference (33) of the revised paper). The quadrant scan approach used by the authors in that paper is similar to our approach of detecting transitions based on the community structure of recurrence networks. However, the quantifier of the ‘block structure’ of the recurrence matrix used in [3] is not the same as the hypothesis-based p -value that we use. Consider the schematic recurrence matrix (at some position k of the sliding window used in our analysis) shown in Figure 1 of this response document, where there is an abrupt transition at $T = 100$. Using the notation from Rapp et al.’s study, we label the four quadrants as A, B, C, and D respectively. Denoting the number of 1’s in the four quadrants by N_A , N_B , N_C , and N_D respectively, the *within-community link fraction* defined in our approach is (cf. Equation 15 of the revised paper),

$$s^k = \frac{N_B + N_C}{N_A + N_B + N_C + N_D}. \quad (1)$$

The denominator on the right-hand side equals the total number of links in the recurrence network, i.e., the link density of the network. This is different from the quantifier used in the Rapp et al. study,

$$s^k = \frac{\left(\frac{N_B + N_C}{N_{NB} + N_{NC}} \right)}{\left(\frac{N_B + N_C}{N_{NB} + N_{NC}} \right) + \left(\frac{N_A + N_D}{N_{NA} + N_{ND}} \right)}, \quad (2)$$

where N_{NA} , N_{NB} , N_{NC} , and N_{ND} are the total number of nodes in quadrants A, B, C, and D respectively.

We would like to emphasise that, despite the similarities, our approach provides a more thorough, hypothesis-based approach to detecting a potential transition at the midpoint of a chosen time period, the particulars of which has not been discussed, to the best of our knowledge, in any previous study.

9. (a) ***how are the network communities detected?***

The fundamental idea behind our transition detection method is to test whether or not an observed 2-community structure in a recurrence network is likely to have occurred due to chance, given the link distribution of the network. This should not be confused with the more standard ‘community detection’ approaches where the goal is to reveal an underlying community structure from the observed distribution of links. In such an approach, the nodes of the network are not ordered and can thus be grouped and regrouped in various ways in order to search for the most optimal partition that uncovers the underlying community structure.

¹https://www.researchgate.net/publication/274905301_Hierarchical_Transition_Chronometries_in_the_Human_Central_Nervous_System

Figure 1: **Quadrant scan method of Rapp et al [3]**. The schematic recurrence matrix shown here contains an abrupt transition at $T = 100$. The four quadrants created by drawing vertical and horizontal lines at the time point of transition are labeled A, B, C, and D respectively in accordance with the notation in the Rapp et al. [3] study. The relative fraction of black points in the above plot (i.e., the number of 1’s) in each quadrant is used to identify the block-like structure that indicates the abrupt transition. However, the precise mathematical form of the quantifier differs between that of Rapp et al. and our current approach.

In our approach however, for each time window, we utilise the fact that the nodes of the recurrence network are ordered because of their corresponding time stamps and ask the question: If we divide the network at the midpoint into two sub-networks, would these two sub-networks correspond to communities of the overall network for that time window? Therefore, at each position of the sliding window, we divide the recurrence network covered in that window into exactly two parts, and then check if the resulting ‘communities’ are likely to have occurred by chance, modeled here as the ensemble of random networks with the same link distribution as the observed one. This is discussed in detail in the revised paper in the Methods subsection “Detecting abrupt transitions using recurrence network community structure”.

(b) *why use this method?*

In order to detect abrupt transitions in the given dataset, it suffices to move a sliding window over the dataset and identify if there exists a transition at the midpoint of each position of the window. Therefore, for each position of the window, we can simply divide the network in two parts at the midpoint, and check if the two parts correspond to communities that are not likely to have occurred due to chance, given the link distribution.

This approach offers two advantages: it is computationally faster than typical com-

munity detection approaches, and it also offers us a p -value against which we can choose to accept or fail to accept the null hypothesis that the 2-community structure could have been obtained by a random network with the same degree sequence.

(c) *what effect does alternative choices have?*

As explained in our response to the previous two comments (9(a), 9(b)), traditional community detection approaches are not suitable for the task of transition detection at the midpoint of a chosen time window. We feel that the question of considering alternative community detection approaches is not relevant here.

10. *Ditto - the choice of the sliding window?*

As mentioned in our response to comment (7), the choice of the sliding window is determined by the timescales at which we wish to detect transitions. In the finance example we focus on monthly timescales, whereas in the ENSO example, we focus on interannual timescales, and in the paleoclimate example we focus on centennial timescales. Our results and inferences are robust to small changes in the sizes of the sliding window, i.e., our inferences are not affected by changing the window size in small amounts.

11. *is the 100 time points in Figure 2 equal to 50 data?*

Yes. However, we have revised the synthetic example and removed the step where we sub-sample the synthetically generated time series. The sub-sampling step had no bearing to the main purpose of the example and only complicated the description of the method, so we chose to omit it. In the revised version, we still use a sliding window of 100 time points for the new synthetic example, which equals 100 time units of the data.

12. *the paper should be carefully proof-read (the authors refer to it as a “letter” on page 2, and on page 7 use the awkward phrase “we have on our hands. . .”*

We thank the reviewer for pointing this out and for the suggestion to get the manuscript properly proof-read. We have revised the text at the points mentioned above and have got it proof-read by individuals other than the authors.

13. *the scale of the colour encoding of the the sliding pdfs in the supplementary information is not particularly helpful - apart from the paleo climate proxies, they all look just like time series on a yellow background.*

We agree with the reviewer that especially in the case of the financial datasets, the sequence of probability densities look just like time series on a yellow background. This is because the $\rho_t^X(x)$ series in the financial case is just a constant value between the daily minimum and maximum (which, in the normalized density plots given in the SI is just 1) and the probability density is zero everywhere else. Compared to the overall drift of the datasets in the entire timespan considered, the daily range of values are very small, which results in these densities appearing almost as point-like objects in the plot.

We have tried other color schemes, as well as logarithmic scales for these plots but these alternatives do not help to improve the visualisation as compared to its current state. To illustrate this point, the BSE SENSEX100 dataset is shown here (Figure 2) with a logarithmically scaled colorbar.

Figure 2: **Logarithmic color scale for probability density series.** We reproduce here Supplementary Figure 1a, but with the color scheme now scaled logarithmically. Even with this modification, the sequence of probability densities looks almost like a time series. This is especially true for the financial dataset, where the possible range of values within one day is quite small compared to the overall drift during the time period of study. However, we stress that this does not undermine the applicability of our approach, which incorporates our inherent ignorance about the intra-day variability of the stock indices.

Reviewer #2

1. *In the paper the authors consider the problem of modeling time series when there are, using the language of the paper, abrupt transitions in the dynamics of the underlying process. The provided real-world examples of such time series include financial stock indices (DAX, Nasdaq-100 and BSE SENSEX are used), sea surface temperatures anomalies from the Niño 3.4 region and paleoclimatic proxies, all of which are described in detail in the paper.*

The main contribution of the paper is the proposal of modeling time series by considering underlying probability densities rather than the observables themselves. The authors highlight the application of their method to paleoclimate time series analysis and claim that this is the first time a “uncertainty-aware” framework is applied in the setting of paleoclimate proxy records.

We thank the reviewer for the summary of our work. The main proposal of our paper was to put forward an alternative ‘representation’ of time series. We would caution against the use of the word ‘modeling’ in this context as it might imply additional connotations not relevant to our work.

2. *The proposed method for modeling a time series, supposedly taking into account what the authors refer to as time series uncertainties, is as follows: From an observed time series $\{x_t\}_{t=1}^n$, the authors model a corresponding sequence of probability densities $\{\varrho_t(x)\}_{t=1}^n$; for each t , $\varrho_t(x)$ is supposed to represent a probability density from which the observed x_t can be viewed as an outcome (more specifically, the density of a random variable for which x_t is viewed as an outcome).*

We would like to clarify that we do not necessarily ‘model’ a probability density series

based on an observed time series. We propose to use the sequence of PDFs $\{\varrho_t^X\}_{t=1}^N$ as an alternative representation of the time evolution of the state \mathcal{X} of the system, made accessible to us by a noisy measurement process which gives us the observable X . The state \mathcal{X} is not observable directly and the PDF ϱ_t^X encodes our partial knowledge about \mathcal{X} since the measurement process introduces a degree of uncertainty, which may arise due to measurement imprecision or also due to spatio-temporal fluctuations. If by x_t , the reviewer implies a traditional time series, then we have to clarify that we do not interpret ϱ_t^X as a probability density from which the observed x_t can be viewed as an outcome. In fact, in most cases, such as in the Niño 3.4 example, the traditional time series x_t (which in this case would be the Niño 3.4 index time series) would correspond to the mean value of the observable taken with respect to the PDFs used in our approach. We would like to stress that the PDFs used in our study encodes our subjective beliefs about the likely state of the system \mathcal{X} at a chosen time $T = t$.

We have stated this terminology and the interpretation of the PDFs that we use in our analysis in the revised version of the manuscript (lines 60–72).

3. *Moreover, abrupt transitions are detected by conducting a so-called recurrence analysis. The densities $\varrho_t(x)$ are used to estimate the probability of recurrence (in an ε -neighborhood sense) for all time points, resulting in an estimated $n \times n$ -matrix \hat{A} of such probabilities. Interpreting this matrix as the adjacency matrix of a network, the authors then characterize abrupt transitions by the presence of so-called communities in this network; communities are characterized by having higher similarity within themselves than the rest of the time series, indicating time intervals with distinct dynamics.*

We agree with the reviewer’s summary of this part of our approach. As mentioned in our response (9) to Reviewer#1, we would like to reiterate that we do not ‘detect’ community structure but test (statistically) if a partition of the network at its midpoint results in two highly unlikely non-random communities (cf. response (9) to Reviewer#1 above for a more detailed explanation).

4. *The novelty of the paper is stated (both implicitly and explicitly) to be (i) the modeling of a sequence of probability densities based on an observed time series, (ii) the combination of (i) and a recurrence network analysis, and (iii) applications to paleoclimate time series. As my expertise covers mainly the modeling part this review will focus on that part of the paper.*

Interpreting ‘modeling’ as ‘representation of’ in the above comment, we agree with the reviewer’s apt summary of the novelty of our paper. We also thank the reviewer for the detailed, in-depth critique of our work. We respond to the issues raised below one-by-one.

5. (a) *Regarding the modeling step (i), the section “Constructing $\varrho_t(x)$ from measurements” is at times confusing. There are inconsistencies, at least in the language used, in whether quantities are considered as outcomes of a stochastic process or as deterministic.*

We apologise for the lack of clarity in this subsection. We have rewritten major portions of this subsection and hope that the revised version offers more clarity than before. Moreover, we have completely rewritten the description of the synthetic

example making the distinction between the deterministic and stochastic components clearer.

- (b) *For the daily stock index data, the use of a uniform distribution over the interval $[x_{lo}(t), x_{hi}(t)]$, taking into consideration that financial data typically exhibits heavy tails.*

We agree that our assumption of a uniform distribution over the interval $[x_{lo}(t), x_{hi}(t)]$ to describe the distribution of the stock index values within a day $T = t$ does not take into account that such data typically have heavy, non-Gaussian tails [4]. However, we make this choice based on two considerations. First, since we do not have access to the actual intra-day values of the stock indices (such datasets are typically proprietary and not freely available), we assume a uniform distribution as a first-order approximation of our ignorance. Second, the choice of the actual form of the distribution is unrelated to the main goal of our paper, which is to show the possibility of utilising the sequence of PDFs $\{\varrho_t^X\}_{t=1}^N$ as a viable time series representation and to be able to detect abrupt transitions using this representation.

- (c) *For the paleoclimatic proxy records, in equation (3) the quantities $w_t(z_j^x)$ are not defined in the paper.*

We are sorry for this omission. We have included the definition appropriately in the text in the revised version.

- (d) *In general, it is not obvious to me how much is added to the modeling by considering densities rather than a model for the observed values, since you still have to make a choice for what type of density to use for a given time series (as opposed to the dependence structure or similar).*

We feel that there was a misunderstanding in communicating the main goal of our paper, which was to demonstrate that the PDF sequence $\{\varrho_t^X\}_{t=1}^N$ is a viable new representation for time series inherently geared to handle uncertainties. The specific details of how the PDF sequences are obtained can vary drastically from one example to another, and this is not the main focus of our paper. We want to stress that even though we made a choice to ‘model’ the intraday variability of the stock index data using a uniform distribution, the densities obtained in the ENSO and the paleoclimatic examples were obtained empirically and determined using nonparametric methods. This does not impact the validity or the efficacy of the method put forward to detect abrupt transitions. In fact, it is a strength of our approach that we are able to detect transitions in different kinds of datasets, even when the initial $\{\varrho_t^X\}_{t=1}^N$ sequences in each example are obtained through different approaches and had different functional forms.

In the revised version, we have tried to clearly state the main focus of our paper at the outset (cf. lines 83–89).

- (e) *Moreover, the authors should be more specific in addressing the aim of their method: Is it primarily for prediction, for inference etc.? This is never mentioned in the paper, although it seems as if it is mainly inference that is intended*

We have added the following lines in the introductory part of the paper in the revised version (cf. lines 83–89):

“The PDFs $\{\varrho_t^X\}_{t=1}^N$ can be estimated in different ways depending on the system under study and the nature of the measurements. In this paper, we will present three

real-world examples from considerably distinct disciplines and the PDF sequences in each of these is constructed in a different way. Irrespective of how the PDFs $\{\varrho_t^X\}_{t=1}^N$ are obtained, the main goal of our paper is to demonstrate that $\{\varrho_t^X\}_{t=1}^N$ is a viable representation for time series datasets, potentially useful in a wide range of real-world applications, and to show how we can detect abrupt transitions in time series with uncertainties with the help of the PDF sequence.”

6. (a) ***For the recurrence analysis, it is again quite difficult to follow the description of the method. Especially since the authors are not clear on what are random variables, constants, distributions etc (see comments at the end of this review).***

A similar point was raised by Reviewer #1 with regard to the description in the subsection “Network of recurrence probabilities”. We have revised the text in this subsection and have included a more detailed, and hopefully more comprehensible description of our recurrence analysis approach (cf. response (3) to Reviewer#1 above for a more detailed explanation).

- (b) ***In the description of how to detect communities from the estimated adjacency matrix, it is not clear what the null hypothesis and involved random variables are.***

We apologise for the ambiguous description of the community structure estimation process. We have extensively re-written the subsection “Detecting abrupt transitions using recurrence network community structure” and hope that the revised version offers more clarity. In the revised version, we have defined the within-community link fraction for a given position k of the sliding window used in our analysis (Equation 15). Moreover, in light of some issues raised by Reviewer #1, we state in the revised text (lines 353–358) that we do not do a community detection in the traditional sense, but test whether the two subnetworks before and after the midpoint of the window form highly unlikely non-random communities.

We would like to mention that we had stated the null hypothesis in the earlier version as well.

- (c) ***What does it mean to “specify the communities before we measure S ”?***

This phrase meant that we need to specify the partitions of the network (i.e. ‘specify’ potential communities to calculate S). However, we have removed this phrase in the revised version.

- (d) ***Moreover, the sampling from the degree-configuration model could be a computationally costly task for large samples (a large number of nodes). How is the sampling done, using a vanilla-type algorithm?***

The network sizes used in sampling the degree configuration model in the analysis correspond to the window sizes (in number of time points) which are not large. As stated in the text, the window sizes are: (i) synthetic example, 100 time points, (ii) financial datasets, 60 time points, (iii) SST data, 30 time points, and (iv) paleoclimatic datasets, 100 time points.

We use the Python Igraph package to carry out the network-related computations. In particular, we use the `Degree_Sequence` routine to sample random networks with a prescribed degree sequence (see²).

²http://igraph.org/python/doc/igraph.GraphBase-class.html#Degree_Sequence

Figure 3: **Sampling random networks with given degree sequence.** We consider a randomly chosen window of 100 time points from the synthetic example presented in the paper, and use the Python Igraph `Degree_Sequence` routine to sample random networks with the degree sequence obtained in the chosen window, but for different sizes of the sample. The figure clearly shows that the degree distribution (represented here by the different percentiles) is reproduced correctly for all sample sizes.

(e) *How do the authors check that sampling has converged to something close to the true distribution?*

In order to verify this, we varied the sample size, i.e., we changed the number of random networks sampled from one particular sliding window (chosen at random) of 100 time points from the synthetic example dataset in the paper. The number of random network samples were changed from 10 to 10000. In each case various percentiles of the within-community link fraction are obtained and plotted along with their standard deviation (Figure 3). The results show that the within-community link fraction converges fairly well with an increase of the ensemble size. Almost the entire distribution (between the 5-th and 95-th percentiles) is confined to an interval of ca. 0.05 (i.e., link fraction lies between 0.48 and 0.52) on the Y-axis in Figure 3. Since all of our significant windows have extremely low p -values (cf. Figures 2–5 of our paper), we contend that our results are robust to small fluctuations in the sampling of the within-community link fraction from degree-sequence constrained random networks.

7. (a) *Although the report may seem negative, I am interested in the authors ideas and the proposed methodology. However, for it to warrant publication in a journal like this it is my opinion that the mathematical level must be raised and the overall presentation improved.*

After incorporating the helpful recommendations of both reviewers, we feel that

the quality of our manuscript has improved considerably. The description of the methods used as well as the mathematical notations both have been revised to give more clarity.

- (b) *Alternatively, the authors could state early on that the modeling in the paleoclimate-setting is the main focus and put the emphasis on those results; for financial time series there are already ways of taking into account shifts in the underlying dynamic and a focus on such time series would then warrant a more in-depth comparison.*

We have now added a phrase in the abstract which makes it clear that the primary focal application in our paper is the paleoclimatic example, where our approach has a lot of potential impact.

“Our approach is particularly relevant for the paleoclimate proxies, we provide for the first time a clear, ‘uncertainty-aware’ framework that validates the hypothesis that ice rafting events in the North Atlantic, known as Bond Events (BEs), are synchronous with a weakening of the Asian summer monsoon (ASM).”

- (c) *As of now, I read the paper as primarily wanting to introduce a methodology, which is poorly described. The authors should also put more effort into comparing their proposed method to other ways of detecting changes in the dynamic of a time series (see the vast literature on change-point-detection) and clearly stating what the improvements are.*

We have substantially revised the text of the manuscript, paying more attention to terminology and the description of the methods.

A comparison of our approach to existing change-point detection methods would be inherently flawed as our approach is defined on the series of PDFs $\{\varrho_t^X\}_{t=1}^N$ and all existing methods are defined on the point-like time series $\{x_t\}_{t=1}^N$. We would like to stress that even if there exists a change-point detection approach that performs better at identifying transitions, it will not be able to incorporate the time series uncertainties that are inherently taken care of in our framework. This is well illustrated in the case of the ‘4.2k event’ that was not detected in the Qunf cave due to the time series uncertainties. Any method which attempts to identify transitions based solely on a point-like estimate of the time evolution of the system will not be able to elucidate the link with initial dataset uncertainties and the final results, as evident in our approach.

In the revised version, we have added references to change-point detection approaches for traditional time series and highlight the lack of treatment of uncertainties in the change-point detection literature at the very outset of the paper (lines 43–46):

“Although there exists a vast literature on various kinds of ‘change-point detection’ methods that seek to address this question [5–9], most of these approaches tend to simplify the nature of uncertainties in the data in exchange for analytical tractability, thereby influencing whether or not a transition is detected in the time series.”

- (d) *Some (additional) relevant questions are:*
- i. *How does one choose what type of density to use in a real-world example?*

We note that Reviewer #1 had raised a similar question regarding the choice of the link density (point (7) of our responses to Reviewer #1). The choice of the link density is motivated by the timescales at which we wish to detect transitions in each dataset.

In all of the examples, the link densities are determined by the timescales of communities that we wish to use in order to detect transitions. This is because the transitions detected in our study are determined by: (i) the size of the sliding window (larger size of window would detect transitions that separate longer periods of similar dynamics), and (ii) the link density of the overall recurrence network. A network of higher link density will lead to a ‘denser’ recurrence matrix as compared to one with a lower link density. A pair of adjacent communities that were detectable at lower link densities might get ‘filled up’ to form a single community at a higher link density, which would cover up the transition that existed between the two at lower link densities. In the finance example we focus on monthly timescales, whereas in the ENSO example, we focus on decadal timescales, and in the paleoclimate example we focus on centennial timescales. Moreover, we would like to point out that in all of these cases, the results as well as all subsequent inferences are robust to small changes in the link density, i.e., our inferences are not affected by changing the link density in small amounts.

ii. ***Are there any performance guarantees or diagnostics that can be used?***

The issue of parameter selection in recurrence plot analysis (such as the recurrence rate/link density, recurrence threshold, choice of embedding, etc.) has been raised in several studies, which also offer guidelines and recommendations on how to avoid spurious/non-robust results. We refer to the studies by Schinkel et al. [10], Marwan [11], and Eroglu et al. [12] for a more detailed discussion on this issue.

8. ***With the above motivation I recommend that the paper be rejected but the authors invited to submit a new version of the paper after a major revision. However, as I am less familiar with the mathematical standard of the journal, I leave it to the AE to decide whether this is indeed a relevant argument.***

I end this review with some minor comments on the text.

(a) ***l. 163) x_0 a stochastic process and thus referring to it as a mean is not proper terminology.***

We have rewritten the description of the synthetic example in the new version, and this mistake is avoided in the revised text (lines 221–237).

(b) ***l. 167) x_0 has a stochastic process, hence it is a poor choice of words to refer to the $x_s(t)$ as “mean(s)” - this suggests a deterministic quantity (see comment on l. 163).***

We have changed the language extensively in this part of the text (lines 221–237).

(c) ***l. 195) The notation “ $\varrho_t(x) \sim \mathcal{N}(x_s(t), \sigma_s)$ ” typically signifies that the former is a random variable distributed according to the latter. Change to something like “ $\varrho_t(x)$ is taken as the density of a $\mathcal{N}(x_s(t), \sigma_s)$ -distribution”.***

This was a notational error in the earlier version of our paper. In the revised version, we write this as (cf. lines 233–235):

“Finally, in order to simulate noisy measurements, we take $\tilde{x}_0^i(t)$ as the mean of a normal distribution with standard deviation equal to the measurement error 1.5 such that our final sampled time series $x_s^i(t)$ is: $x_s^i(t) \sim \mathcal{N}(\tilde{x}_0^i(t), 1.5)$ ”

- (d) *l. 224) $P_{Z_{ij}}$ is here referred to as a distribution but later (l. 226) as a scalar value: $P_{Z_{ij}} \in [m_{ij}, M_{ij}]$. Be clear on what you mean. For example, is the interval the support of the distribution?*

We apologise for this error. We meant it to be understood as the CDF, which was not clear from the text. In the revised version, we clearly define it as the CDF of Z_{ij} along with Equation 7. Moreover, we clearly state that the bounds M_{ij} and m_{ij} are bounds on the values taken by the CDF for all possible values of z_{ij} (cf. lines 309–311 and 319–322).

- (e) *l. 231) Q_{ij} is a probability but here treated as a random variable. Are the authors really considering the conditional probabilities given the observed values of X ?*

Yes, we treat $Q_{ij}(\varepsilon)$ as a random variable even though it is a probability because we do not have the means to exactly estimate the probability. This is made clear in Equation 5 of the revised version of the manuscript.

Also, we do not consider $Q_{ij}(\varepsilon)$ by conditioning on an observed value of X . We would like to note that we are a bit unclear what the reviewer intended to say in this question.

- (f) *Eq. (5)) Define also $\varrho(\mathbf{A}_{ij} = 1|q_{ij})$. Readily apparent from the context but should still be explicitly defined (since ϱ is used for other things as well).*

We have defined this and other probabilities used in the derivation of the adjacency matrix estimator $\hat{\mathbf{A}}$ in greater detail in the revised version.

- (g) *l. 236) Do you mean uniformly here, rather than “symmetrically”? If not, how is the probability mentioned on l. 237 obtained?*

Yes, we mean “symmetrically” here, and not “uniformly”. This is because we end up with a quantity of interest $Prob(\mathbf{A}_{ij} = 1)$ as the expectation value of a random variable Q_{ij} in a finite interval. In such a case, under the assumption that Q_{ij} is distributed *symmetrically* around its mean, the expectation value has to be the average of the end points of the interval.

Although we would get the same result for a uniform distribution, it is only because a uniformly distributed random variable is distributed symmetrically around its mean.

- (h) *l. 258) The authors mentions the networks “degree-sequence” without defining this quantity. Readers specialized in either time series or the applied areas addressed in the paper are not necessarily familiar with graph-theoretic concepts.*

In the revised version, we define the concept of the *degree-sequence* at the occurrence of its first usage in the text (line 373 of the revised paper).

References

- [1] Marwan, N., et al., Complex network approach for recurrence analysis of time series. *Phys. Lett. A* **373**, 4246–4254 (2009).
- [2] Williamson, R. C. & Downs, T. Probabilistic arithmetic. I. Numerical methods for calculating convolutions and dependency bounds. *Int. J. Approx. Reason.* **4**, 89–158 (1990).
- [3] Rapp, P. E., Darmon, D. M. & Cellucci, C. J. Hierarchical Transition Chronometries in the Human Central Nervous System. *Proceedings of 2013 International Symposium on Nonlinear Theory and its Applications* **2**, 286–289 (2014).
- [4] Gerig, A. Vicente, J. & Fuentes, M. A. Model for non-Gaussian intraday stock returns.. *Physical Review E* (2009).
- [5] Wu, B. & Chen, M.-H. Use of fuzzy statistical technique in change periods detection of nonlinear time series. *Applied Mathematics and Computation* **99**, 241–254 (1999).
- [6] Ray, B. K. & Tsay, R. S. Bayesian methods for change-point detection in long-range dependent processes. *Journal of Time Series Analysis* **23**, 687–705 (2002).
- [7] Aue, A. & Horváth, L. Structural breaks in time series. *Journal of Time Series Analysis* **34**, 1–16 (2012).
- [8] Cho, H. & Fryzlewicz, P. Multiple-change-point detection for high dimensional time series via sparsified binary segmentation. *Journal of the Royal Statistical Society: Series B* **77**, 475–507 (2014).
- [9] Aminikhanghanhi, S. & Cook, D. J. A survey of methods for time series change point detection. *Knowledge and Information Systems* **51**, 339–367 (2017).
- [10] Schinkel, S. Dimigen, O. & Marwan, N. Selection of recurrence threshold for signal detection.. *European Physical Journal Special Topics* **164**, 45–53 (2008).
- [11] Marwan, N. How to avoid potential pitfalls in recurrence plot based data analysis.. *International Journal of Bifurcation and Chaos* **21**, (2008).
- [12] Eroglu, D. et al. Finding recurrence networks’ threshold adaptively for a specific time series.. *Nonlinear Processes in Geophysics* **21**, 1085–1092 (2014).

REVIEWERS' COMMENTS:

Reviewer #1 (Remarks to the Author):

I am satisfied with the revisions and support publication in Nature Communications.

Reviewer #2 (Remarks to the Author):

I want to commend the authors for a well-written, well-thought-out response to the referee reports. The changes have, in my opinion, greatly improved the paper and I see no reason why it should not be published.

Response to reviewers' comments
NCOMMS-17-13021-A
Abrupt transitions in time series with uncertainties

Bedartha Goswami, Niklas Boers, Aljoscha Rheinwalt,
Norbert Marwan, Jobst Heitzig, Sebastian F. M. Breitenbach, and Jürgen Kurths

Reviewer #1

- *I am satisfied with the revisions and support publication in Nature Communications.*

We thank the reviewer for this response and for supporting the publication of our paper in *Nature Communications*.

Reviewer #2

- *I want to commend the authors for a well-written, well-thought-out response to the referee reports. The changes have, in my opinion, greatly improved the paper and I see no reason why it should not be published.*

We thank the reviewer for the positive comment and for supporting the publication of the paper in *Nature Communications*. We agree wholeheartedly with the reviewer that the quality of our paper has increased substantially during the peer review.